# SpikeInterface, a unified framework for spike sorting

**Alessio P Buccino[1,2†]\*, Cole L Hurwitz[3†], Samuel Garcia[4], Jeremy Magland[5], Joshua H Siegle[6], Roger Hurwitz[7], Matthias H Hennig[3]**

[1]Department of Biosystems Science and Engineering, ETH Zurich, Zürich, Switzerland; [2]Centre for Integrative Neuroplasticity (CINPLA), University of Oslo, Oslo, Norway; [3]School of Informatics, University of Edinburgh, Edinburgh, United Kingdom; [4]Centre de Recherche en Neuroscience de Lyon, CNRS, Lyon, France; [5]Flatiron Institute, New York, United States; [6]Allen Institute for Brain Science, Seattle, United States; [7]Independent Researcher, Portland, United States

**Abstract** Much development has been directed toward improving the performance and automation of spike sorting. This continuous development, while essential, has contributed to an over-saturation of new, incompatible tools that hinders rigorous benchmarking and complicates reproducible analysis. To address these limitations, we developed SpikeInterface, a Python framework designed to unify preexisting spike sorting technologies into a single codebase and to facilitate straightforward comparison and adoption of different approaches. With a few lines of code, researchers can reproducibly run, compare, and benchmark most modern spike sorting algorithms; pre-process, post-process, and visualize extracellular datasets; validate, curate, and export sorting outputs; and more. In this paper, we provide an overview of SpikeInterface and, with applications to real and simulated datasets, demonstrate how it can be utilized to reduce the burden of manual curation and to more comprehensively benchmark automated spike sorters.

**\*For correspondence:**
alessio.buccino@bsse.ethz.ch

[†]These authors contributed equally to this work

**Competing interests:** The authors declare that no competing interests exist.

## Introduction

Extracellular recording is an indispensable tool in neuroscience for probing how single neurons and populations of neurons encode and transmit information. When analyzing extracellular recordings, most researchers are interested in the spiking activity of individual neurons, which must be extracted from the raw voltage traces through a process called *spike sorting*. Many laboratories perform spike sorting using fully manual techniques (e.g. XClust [*Mucha, 1995*], SimpleClust [*Voigts, 2012*], Plexon Offline Sorter [*Plexon, 2020*]), but such approaches are nearly impossible to standardize due to inherent operator bias (*Wood et al., 2004*). To alleviate this issue, spike sorting has seen decades of algorithmic and software improvements to increase both the accuracy and automation of the process (*Rey et al., 2015*). This progress has accelerated in the past few years as high-density devices (*Eversmann et al., 2003*; *Berdondini et al., 2005*; *Frey et al., 2010*; *Ballini et al., 2014*; *Müller et al., 2015*; *Yuan et al., 2016*; *Lopez et al., 2016*; *Jun et al., 2017a*; *Dimitriadis et al., 2018*; *Angotzi et al., 2019*), capable of recording from hundreds to thousands of neurons simultaneously have made manual intervention impractical, increasing the demand for both accurate and scalable spike sorting algorithms (*Rossant et al., 2016*; *Pachitariu et al., 2016*; *Lee et al., 2017*; *Chung et al., 2017*; *Yger et al., 2018*; *Hilgen et al., 2017*; *Jun et al., 2017b*; *Diggelmann et al., 2018*).

Despite the development and widespread use of automatic spike sorters, there still exist no clear standards for how spike sorting should be performed or evaluated (*Rey et al., 2015*; *Barnett et al., 2016*; *Carlson and Carin, 2019*; *Magland et al., 2020*). Research labs that are beginning to experiment with high-density extracellular recordings have to choose from a multitude of spike sorters,

data processing algorithms, file formats, and curation tools just to analyze their first recording. As trying out multiple spike sorting pipelines is time-consuming and technically challenging, many labs choose one and stick to it as their de facto solution (*Magland et al., 2020*). This has led to a fragmented software ecosystem which challenges reproducibility, benchmarking, and collaboration among different research labs.

Previous work to standardize the field has focused on developing open-source frameworks that make extracellular analysis and spike sorting more accessible (*Egert et al., 2002*; *Bonomini et al., 2005*; *Hazan et al., 2006*; *Garcia and Fourcaud-Trocmé, 2009*; *Goldberg et al., 2009*; *Bokil et al., 2010*; *Xq et al., 2011*; *Bologna et al., 2010*; *Oostenveld et al., 2011*; *Kwon et al., 2012*; *Mahmud et al., 2012*; *Bongard et al., 2014*; *Regalia et al., 2016*; *Zhang et al., 2017*; *Nasiotis et al., 2019a*). While useful tools in their own right, these frameworks only implement a limited suite of spike sorting technologies since their main focus is to provide *entire* extracellular analysis pipelines (spike trains, LFPs, EEG, and more). Moreover, these tools do little to improve the evaluation and comparison of spike sorting performance which is still a relatively unsolved problem in electrophysiology. An exception to this is SpikeForest (*Magland et al., 2020*), a recently developed open-source software suite that benchmarks 10 automated spike sorting algorithms against an extensive database of ground-truth recordings (SpikeForest makes use of SpikeInterface in many of its core capabilities [file IO, preprocessing, spike sorting]). Despite these developments, there exists a need for an up-to-date spike sorting framework that can standardize the usage and evaluation of modern algorithms.

In this paper, we introduce SpikeInterface, the first open-source, Python-based framework exclusively designed to encapsulate all steps in the spike sorting pipeline (we utilize Python as it is open-source, free, and increasingly popular in the neuroscience community; *Muller et al., 2015*; *Gleeson et al., 2017*). The goals of this software framework are five-fold.

1. To increase the accessibility and standardization of modern spike sorting technologies by providing users with a simple application programming interface (API) and graphical user interface (GUI) that exist within a continuously integrated code-base.
2. To make spike sorting pipelines fully reproducible by capturing the entire provenance of the data flow during run time.
3. To make data access and analysis both memory and computation-efficient by utilizing memory-mapping, parallelization, and high-performance computing platforms.
4. To encourage the sharing of datasets, results, and analysis pipelines by providing full compatibility with standardized file formats such as Neurodata Without Borders (NWB) (*Teeters et al., 2015*; *Ruebel et al., 2019*) and the Neuroscience Information Exchange (NIX) Format (*NIX, 2015*).
5. To supply the most comprehensive suite of benchmarking capabilities available for spike sorting in order to guide future usage and development.

In the remainder of this article, we showcase the numerous capabilities of SpikeInterface by performing an in-depth meta-analysis of preexisting spike sorters. This analysis includes quantifying the agreement among six modern spike sorters for dense probe recordings, benchmarking each sorter on ground truth, and introducing a consensus-based technique to potentially improve performance and enable automated curation. Afterwards, we present an overview of the codebase and how its interconnected components can be utilized to build full spike sorting pipelines. Finally, we contrast SpikeInterface with preexisting analysis frameworks and outline future directions.

## Results

In this section, we perform a meta-analysis of six modern spike sorters on real and simulated datasets. This meta-analysis includes quantifying agreement among the sorters, benchmarking each sorter on ground truth, and investigating whether it is possible to combine outputs from multiple spike sorters to improve overall performance and to reduce the burden of manual curation. All analyses are done with `spikeinterface` version 0.10.0 which is available on `PyPI` (https://pypi.org/project/spikeinterface/). The code to perform this analysis and produce all figures can be found at https://spikeinterface.github.io/ which also showcases other experiments performed using SpikeInterface. The datasets are publicly available in NWB format on the DANDI archive (https://gui.dandiarchive.org/#/dandiset/000034/draft).

## Spike sorters show low agreement for the same high-density dataset

The dataset we use in this analysis is a Neuropixels recording from a head-fixed mouse acquired at the Allen Institute for Brain Science (*Siegle et al., 2019a*; *Allen Institute for Brain Science, 2019* dataset ID: 766640955; probe ID: 77359232). The recording has 246 active recording channels (the remaining of the 384 Neuropixels channels were either not inserted in the brain tissue or had a firing rate below 0.1 Hz), and a sampling frequency of 30 kHz. The recording's duration was trimmed to 15 min. The probe records from part of the cortex (V1), the hippocampus (CA1), the dentate gyrus, and the thalamus (LP). During the experiment, the mouse was presented with a variety of visual stimuli while freely running on a rotating disk (for more details see *Siegle et al., 2019a*). An activity map of the probe and a 1 s snippet of the traces on 10 channels are shown in *Figure 1A*. The notebook for reproducing the results for this section and the last section of the Results can be viewed at https://spikeinterface.github.io/blog/ensemble-sorting-of-a-neuropixels-recording.

For this analysis, we select six different spike sorters: HerdingSpikes2 (*Hilgen et al., 2017*), Kilosort2 (*Pachitariu et al., 2018*), IronClust (*Jun et al., 2017b*), SpyKING Circus (*Yger et al., 2018*), Tridesclous (*Garcia and Pouzat, 2015*), and HDSort (*Diggelmann et al., 2018*) (the versions for each spike sorter are as follows: SpyKING Circus==0.9.7, Tridesclous==1.6.0, HerdingSpikes2==0.3.7, IronClust==5.9.8, Kilosort2==GitHub commit 48bf2b81d8ad, HDSort==1.0.1). As most of these algorithms have been tuned rigorously on multiple ground-truth datasets (including the recent large-scale evaluation from *Magland et al., 2020*), we fix their parameters to default values to allow for straightforward comparison. We do not include Klusta (*Rossant et al., 2016*), WaveClus (*Chaure et al., 2018*), Kilosort (*Pachitariu et al., 2016*), or MountainSort4 (*Chung et al., 2017*) in this analysis as Klusta can only handle up to 64 channels, WaveClus is designed for low channel count probes, Kilosort is superseded by Kilosort2, and MountainSort4's latest verion is currently not optimized for high channel counts, scaling quadratically with the number of channels.

In *Figure 1B*, we show the number of units that each of the six sorters output. Immediately, we observe large variability among the sorters, with Tridesclous (TDC) finding the least units (187) and SpyKING Circus (SC) finding the most units (628). HerdingSpikes2 finds 210 units; Kilosort2 finds 446 units; IronClust finds 233 units; and HDSort finds 317 units. From this result, we can see that there is no clear consensus among the sorters on the number of neurons in the recording (without performing extensive manual curation).

Next, we compare the unit spike trains found by each sorter to determine the level of agreement among the different algorithms (see the SpikeComparison Section of the Methods for how this is done). In *Figure 1C*, we visualize the total number of units for which *k* sorters agree (unit agreement is defined as a 50% spike train match; the time window to consider spikes as matching is 0.4 ms). *Figure 1—figure supplement 1* shows spike trains and templates for two sample matched units (one with a higher - 0.97 - and one with a lower agreement - 0.69). Of the 2031 total detected units, all six sorters agree on *just 33 of the units*. This is surprisingly low given the relatively undemanding criteria of a 50% spike train match. We also find that two or more sorters agree on just 263 of the total units. To further break down the disagreement between spike sorters, *Figure 1D* shows the number of units per sorter for which *k* other sorters agree. For most sorters, over 50% of the units that they find do not match with any other sorter (with the exceptions of Ironclust and Tridesclous). For agreed-upon units, around 80% of the agreement scores are 0.8 or higher, indicating that matched units typically have high spike train agreement (*Figure 1—figure supplement 2*).

The analysis performed on this dataset suggests that agreement among spike sorters is startlingly low. To corroborate this finding, we repeat the same analysis using different datasets including a Neuropixels recordings from another lab and an in vitro retinal recording from a planar, high-density array. In both cases, we find similar disagreement among the sorters (*Figure 1—figure supplements 3* and *4*). The notebooks for these analyses can be viewed at https://spikeinterface.github.io/blog/ensemble-sorting-of-a-neuropixels-recording-2/ and https://spikeinterface.github.io/blog/ensemble-sorting-of-a-3brain-biocam-recording-from-a-retina/.

This low agreement raises the following question: how many of the total outputted units actually correspond to real neurons? To explore this question, we turn to simulation where the ground-truth spiking activity is known a priori.

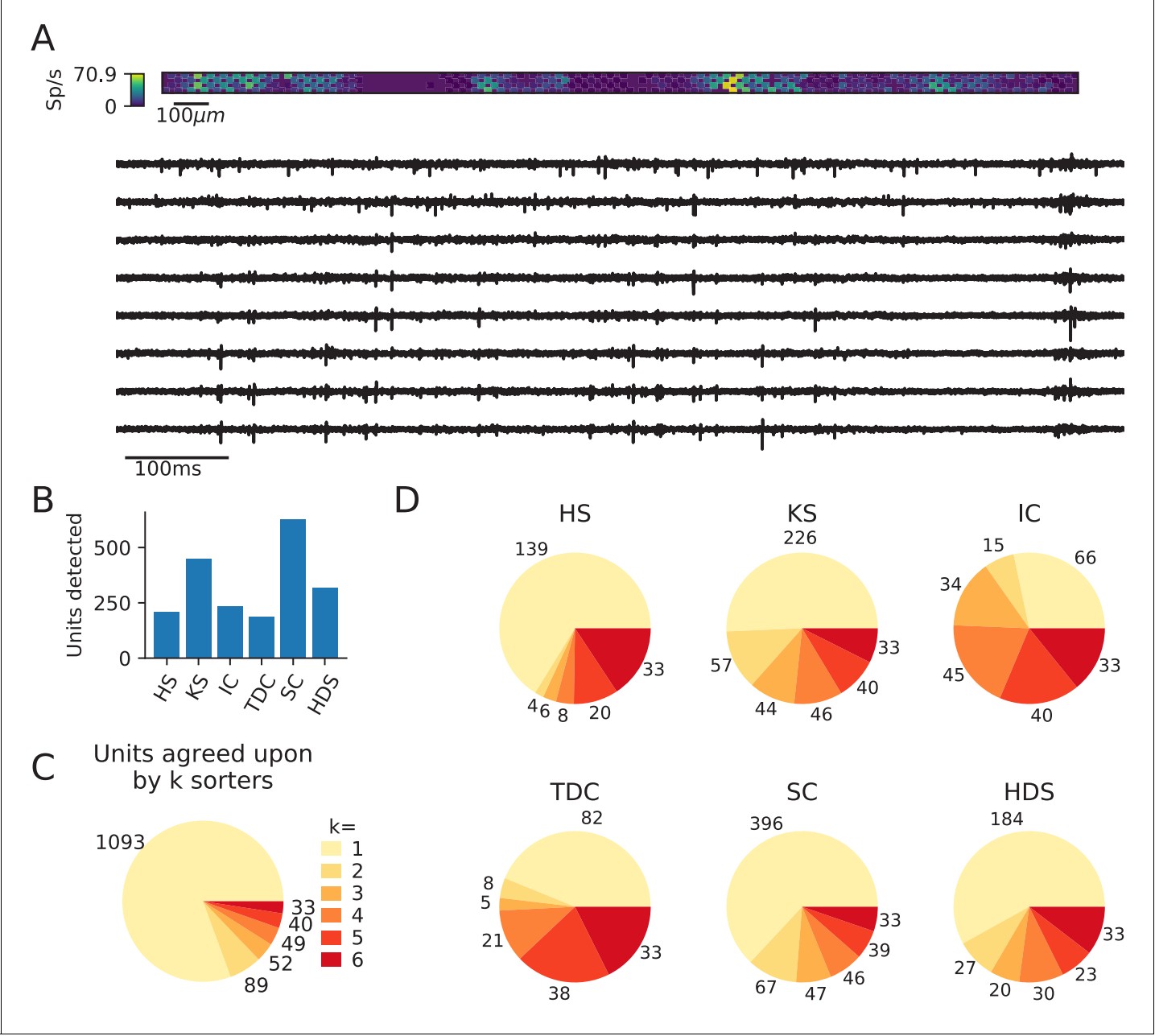

**Figure 1.** Comparison of spike sorters on a real Neuropixels dataset. (A) A visualization of the activity on the Neuropixels array (top, color indicates spike rate estimated on each channel evaluated with threshold detection) and of traces from the Neuropixels recording (below). (B) The number of detected units for each of the six spike sorters (HS = HerdingSpikes2, KS = Kilosort2, IC = IronClust, TDC = Tridesclous, SC = SpyKING Circus, HDS = HDSort). (C) The total number of units for which k sorters agree (unit agreement is defined as 50% spike match). (D) The number of units (per sorter) for which k sorters agree; most sorters find many units that other sorters do not.

The online version of this article includes the following figure supplement(s) for figure 1:

**Figure supplement 1.** Examples of matched units in a Neuropixels recording.

**Figure supplement 2.** Cumulative histogram of agreement scores (above threshold of .5 that defines a match) for the ensemble sorting of the simulated ground-truth dataset.

**Figure supplement 3.** Comparison of spike sorters on a Neuropixels recording.

**Figure supplement 4.** Comparison of spike sorters on a Biocam recording from a mouse retina.

## Evaluating spike sorters on a simulated dataset

In this analysis, we simulate a 10 min Neuropixels recording using the MEArec Python package (*Buccino and Einevoll, 2020*). The recording contains the spiking activity of 250 biophysically detailed neurons (200 excitatory and 50 inhibitory cells from the Neocortical Micro Circuit Portal; *Ramaswamy et al., 2015*; *Markram et al., 2015*) that exhibit independent Poisson firing patterns. The recording also has an additive Gaussian noise with 10 μV standard deviation. A visualization of the simulated activity map and extracellular traces from the Neuropixels probe is shown in *Figure 2A*. A histogram of the signal-to-noise ratios (SNR) for the ground-truth units is shown in *Figure 2B*. The notebook for reproducing the results for this and the next section can be viewed at https://spikeinterface.github.io/blog/ground-truth-comparison-and-ensemble-sorting-of-a-synthetic-neuropixels-recording/.

We run the same six spike sorters on the simulated dataset, keeping the parameters the same as those used on the real Neuropixels dataset. We then utilize SpikeInterface to evaluate each spike sorter on the ground-truth dataset. Afterwards, we repeat the agreement analysis from the previous section to diagnose the low agreement among sorters.

The main result of the ground-truth evaluation is summarized in *Figure 2*. As can be seen in *Figure 2C*, the sorters, again, have a large discrepancy in the number of detected units. The number of detected units range from the 189 units found by Tridesclous to the 458 units found by HDSort. HerdingSpikes2 finds 233 units; Kilosort2 finds 415 units; IronClust finds 283 units; and SpyKING Circus finds 343 units. We again see that there is no clear consensus among the sorters on the number of neurons in the simulated recording.

In *Figure 2D*, the accuracy, precision, and recall of all the ground-truth units are plotted for each spike sorter. Some sorters tend to favor precision over recall while others do the opposite (*Figure 2—figure supplement 1A*). Moreover, the accuracy is modulated by the SNR of the ground-truth units for all spike sorters except Kilosort2 which achieves an almost perfect performance on the low-SNR units (*Figure 2—figure supplement 1B*). While most spike sorters have a wide range of scores for each metric, Kilosort2 attains significantly higher scores than the rest of the spike sorters for most ground-truth units.

*Figure 2E* shows the breakdown of detected units for each spike sorter. Each unit is classified as *well-detected*, *false positive*, *redundant*, and/or *overmerged* by SpikeInterface (the definitions of each unit type can be found in the SpikeComparison Section of the Materials and methods). This plot, interestingly, may shed some light on the remarkable accuracy of Kilosort2. While Kilosort2 has the most well-detected units (245), this comes at the cost of a high percentage of false positive (147) and redundant (21) units (The high-rate of false positive/redundant units persists, but is alleviated, even when using Kilosort2's automated curation step which removes units that have >20% estimated contamination rate [computed from the refractory period violations ]. In that case the number of well-detected units is 241, false positives are 93, and redundant units are 18. In both cases two overmerged units are found). Notably, Tridesclous detects very few false positive/redundant units while still finding many well-detected units. HDSort, on the flip side, finds many more false positive units than any other spike sorter. For a comprehensive comparison of spike sorter performance on both real and simulated datasets, we refer the reader to the related SpikeForest project (https://spikeforest.flatironinstitute.org/) (*Magland et al., 2020*).

## Low-agreement units are mainly false positives

Similarly to the real Neuropixels dataset, we compare the agreement among the different spike sorters on the simulated dataset. Again, we observe a large disagreement among the spike sorting outputs with only 139 units of the 1921 total units (7.24%) being in agreement among all sorters (*Figure 3A*). We can break down the overall agreement by sorter (*Figure 3B*), highlighting that some sorters are more prone to finding low agreement units (HDSort, SpyKING Circus, Kilosort2) than other sorters (HerdingSpikes2, Ironclust, Tridesclous).

Given that we know the ground-truth spiking activity of the simulated recording, we can now investigate whether low-agreement units actually correspond to ground-truth units or if they are falsely detected (false positive) units. In *Figure 3C*, bar plots for each sorter show the number of matched ground-truth units (blue) and false positive units (red) in relation to the ensemble agreement (1 - no agreement, 6 - full agreement). The plots show that (almost) all false positive units are

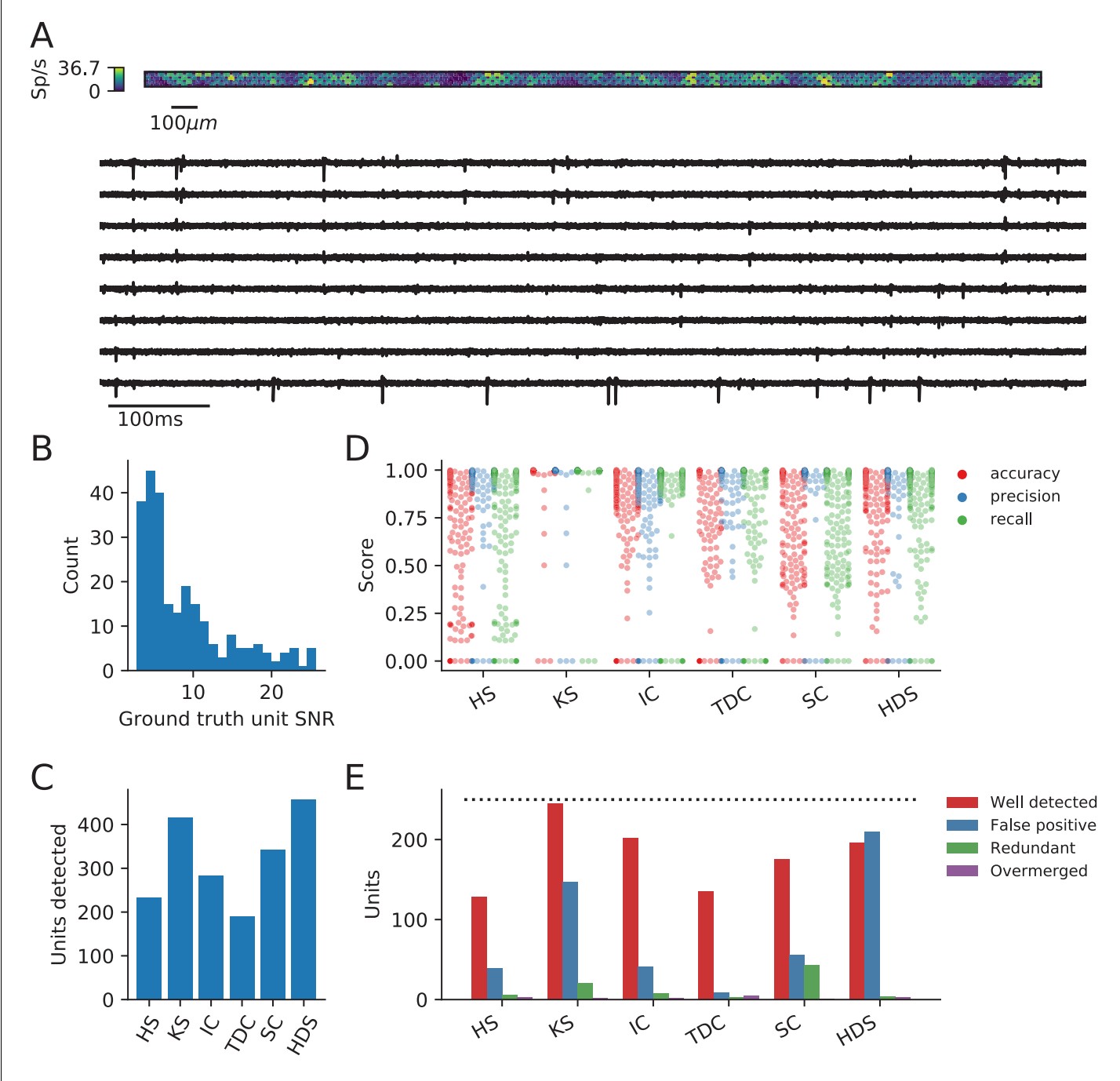

**Figure 2.** Evaluation of spike sorters on a simulated Neuropixels dataset. (**A**) A visualization of the activity on and traces from the simulated Neuropixels recording. (**B**) The signal-to-noise ratios (SNR) for the ground-truth units. (**C**) The number of detected units for each of the six spike sorters (HS = HerdingSpikes2, KS = Kilosort2, IC = IronClust, TDC = Tridesclous, SC = SpyKING Circus, HDS = HDSort). (**D**) The accuracy, precision, and recall of each sorter on the ground-truth units. (**E**) A breakdown of the detected units for each sorter (precise definitions of each unit type can be found in the SpikeComparison Section of the Methods). The horizontal dashed line indicates the number of ground-truth units (250).

The online version of this article includes the following figure supplement(s) for figure 2:

**Figure supplement 1.** Evaluation of spike sorters performance metrics.

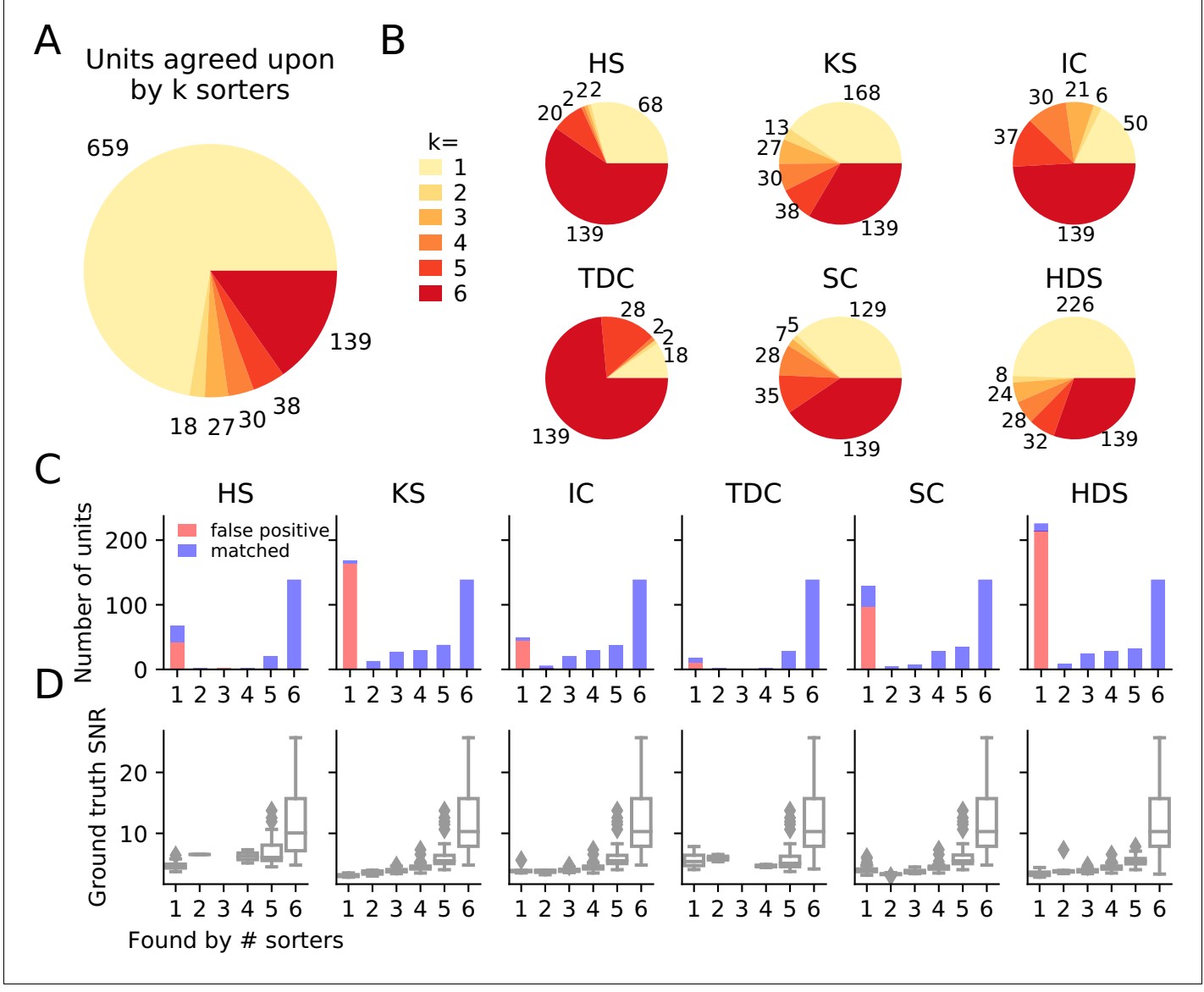

**Figure 3.** Comparison of spike sorters on a simulated Neuropixels dataset. (**A**) The total number of units for which *k* sorters agree (unit agreement is defined as 50% spike match). (**B**) The number of units (per sorter) for which *k* sorters agree; Most sorters find many units that other sorters do not. (HS = HerdingSpikes2, KS = Kilosort2, IC = IronClust, TDC = Tridesclous, SC = SpyKING Circus, HDS = HDSort) (**C**) Number of matched ground-truth units (blue) and false positive units (red) found by each sorter on which *k* sorters agree upon. Most of the false positive units are only found by a single sorter. Number of false positive units found by $k \geq 2$ sorters: HS = 4, KS = 4, IC = 4, SC = 2, TDC = 1, HDS = 2. (**D**) Signal-to-noise ratio (SNR) of ground-truth unit with respect to the number of *k* sorters agreement. Results are split by sorter.

The online version of this article includes the following figure supplement(s) for figure 3:

**Figure supplement 1.** The fractions of predicted false and true positive units from ensembles using different numbers of sorters.

**Figure supplement 2.** The SNR of all units found by Kilosort2 in the ground-truth data separated into those with and without matches in the ground-truth spike trains.

ones that are found by only a single sorter (not matched with any other sorters), while most real units are matched by more than one sorter. We also assessed how well false positive units can be identified using fewer sorters (*Figure 3—figure supplement 1*). This analysis showed that using a pair of sorters is sufficient to isolate almost all false positive units in each sorter, yet when fewer than four sorter outputs are compared, a significant fraction of true positive units found by only one sorter can be wrongly classified as false positives with this approach. For two sorters, the most reliable identification of true positives for this dataset was achieved by combining Kilosort2 and Ironclust (96% and

95% false positive and true positive detection rate, respectively). In *Figure 3D*, we display the signal-to-noise ratio (SNR) as a function of the ensemble agreement. This shows, as expected, that higher SNR units have higher agreement among sorters. In other words, units with a large amplitude (high SNR) are easier to detect and more consistently found by many sorters. Additionally, we tested if SNR can be used to distinguish between false and true positive units, as noise may be wrongly detected as events with low SNR. We found that for Kilosort2's output, which is best matched with ground-truth spike trains, SNR is not a good predictor of false positives (*Figure 3—figure supplement 2*) - many false positives had a high estimated SNR. Taken together, these results suggest that the ensemble agreement among multiple sorters can be used to remove false positive units from each of the sorter outputs or to inform their subsequent manual curation.

## Consensus units highly overlap with manually curated ones

We next investigate the ensemble agreement among the sorters on the real Neuropixels recording presented in *Figure 1*. As there is no ground-truth information in this setting to identify false positives, we turn to manually curated sorting outputs. Two experts (which we will refer to as C1 and C2) manually curate the spike sorting output of Kilosort2 using the Phy software. During this curation step, the two experts label the sorted units as false positives or real units by rejecting, splitting, merging, or accepting units according to spike features (*Rossant and Harris, 2013*).

*Figure 4A* shows the agreement between expert 1 (C1) and expert 2 (C2). While there are some discrepancies (as expected when manually curating spike sorting results; *Wood et al., 2004*), most of the curated units (226 out of 351–64.2%) are agreed upon by both experts. Notably, 174 units found by Kilosort2 are discarded by both experts, indicating a large number of false positive units.

We then compare the output of each of the spike sorters to C1 and C2 and find that, in general, only a small percentage of units outputted by any single sorter is matched to the curated results (*Figure 4*). The highest percentage match is actually IronClust which is surprising given that the initial sorting output was curated from Kilosort2's output (IC $\bigcap$ C1 = 59.83%, IC $\bigcap$ C2 = 61.1%, KS $\bigcap$ C1 = 50.67%, KS $\bigcap$ C2 = 56.25%).

Next, for each sorter, we take all the units that are matched by at least one other sorter (*consensus units*, $k \geq 2$) and all units that are found by only that sorter (*non-consensus units*, $k = 1$). We refer to the consensus units of a sorter as Sorter$_c$ and the non-consensus units of a sorter as Sorter$_{nc}$. In *Figure 4C*, we show the match percentage between consensus units and curated units. The average match percentage is above 70% for all sorters showing that there is a large agreement between the manually curated outputs and the consensus-based output. Kilosort2 has the highest match (KS$_c$ $\bigcap$ C1 = 84.55%, KS$_c$ $\bigcap$ C2 = 89.55%), slightly higher than Ironclust (IC$_c$ $\bigcap$ C1 = 82.63%, IC$_c$ $\bigcap$ C2 = 83.83%). Conversely, the percentage of non-consensus units matched to curated units is very small (*Figure 4D*) for all sorters.

Overall, this analysis suggests that a consensus-based approach to curation could allow for identification of real neurons from spike sorted data. Despite differences among the sorters with respect to the number of detected neurons and the quality of their isolation (as demonstrated by the ground-truth analysis), the consensus-based approach has good agreement with hand-curated data and appears to be less variable as illustrated by the small but significant disagreement between the two curators.

## Materials and methods

### Overview of SpikeInterface

SpikeInterface consists of five main Python packages designed to handle different steps in the spike sorting pipeline: (i) `spikeextractors`, for extracellular recording, sorting output, and probe file I/O; (ii) `spiketoolkit` for low level processing such as pre-processing, post-processing, validation, curation; (iii) `spiketoolkit` for spike sorting algorithms and job launching functionality; (iv) `spikecomparison` for sorter comparison, ground-truth comparison, and ground-truth studies; and (v) `spikewidgets`, for data visualization.

These five packages can be installed and used through the `spikeinterface` metapackage, which contains stable versions of all five packages as internal modules (see *Figure 5*). With these five packages (or our meta-package), users can build, run, and evaluate full spike sorting pipelines in a

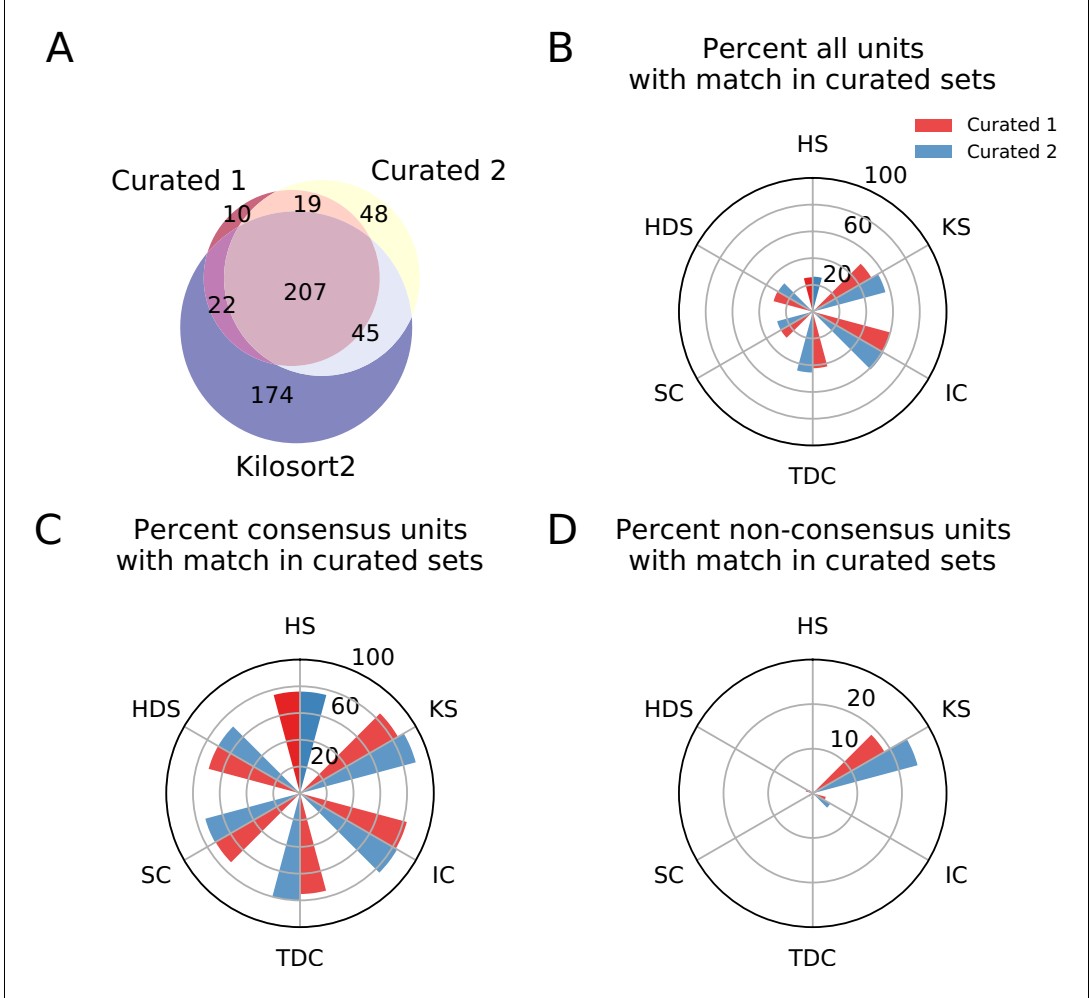

**Figure 4.** Comparison between consensus and manually curated outputs. (**A**) Venn diagram showing the agreement between Curator 1 and 2. 174 units are discarded by both curators from the Kilosort2 output. (**B**) Percent of matched units between the output of each sorter and C1 (red) and C2 (blue). Ironclust has the highest match with both curated datasets. (**C**) Similar to C, but using the consensus units (units agreed upon by at least two sorters - $k \geq 2$). The percent of matching with curated datasets is now above 70% for all sorters, with Kilosort2 having the highest match ($KS_c \cap C1 = 84.55\%$, $KS_c \cap C2 = 89.55\%$), slightly higher than Ironclust ($IC_c \cap C1 = 82.63\%$, $IC_c \cap C2 = 83.83\%$). (**D**) Percent of non-consensus units ($k = 1$) matched to curated datasets. The only significant overlap is between Curator one and Kilosort2, with a percent around 18% ($KS_{nc} \cap C1 = 18.58\%$, $KS_{nc} \cap C2 = 24.34\%$).

reproducible and standardized way. In the following subsections, we present an overview of, and a code snippet for, each package.

## SpikeExtractors

The `spikeextractors` package (https://github.com/SpikeInterface/spikeextractors; *Buccino et al., 2020a*) is designed to alleviate issues of any file format incompatibility within spike sorting without creating additional file formats. To this end, `spikeextractors` contains two core Python objects that can directly and uniformly access all spike sorting related files: the `RecordingExtractor` and the `SortingExtractor`.

The `RecordingExtractor` directly interfaces with an extracellular recording and can query it for four primary pieces of information: (i) the extracellular recorded traces; (ii) the sampling frequency; (iii) the number of samples, or frames, in the recording; and (iv) the channel indices of the recording electrodes. These data are shared across all extracellular recordings allowing for standardized retrieval functions. In addition, a `RecordingExtractor` may store extra information about the recording device as 'channel properties' which are key–value pairs. This includes properties such as

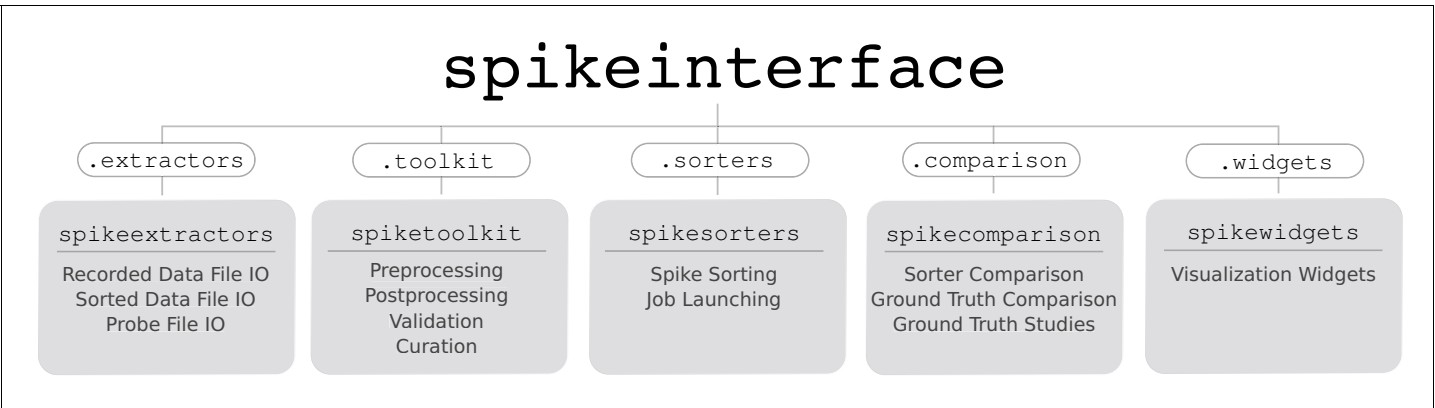

**Figure 5.** Overview of SpikeInterface's Python packages, their different functionalities, and how they can be accessed by our meta-package, spikeinterface.

'location', 'group', and 'gain' which are either provided by certain extracellular file formats, loaded manually by the user, or loaded automatically with our built-in probe file (.prb or .csv) reader. Taken together, the `RecordingExtractor` is an object representation of an extracellular recording and the associated probe configuration.

The `SortingExtractor` directly interfaces with a sorting output and can query it for two primary pieces of information: (i) the unit indices and (ii) the spike train of each unit. Again, these data are shared across all sorting outputs. A `SortingExtractor` may also store extra information about the sorting output as either 'unit properties' or 'unit spike features', key–value pairs which store information about the individual units or the individual spikes of each unit, respectively. This extra information is either loaded from the sorting output, loaded manually by the user, or loaded automatically with built-in post-processing tools (discussed in the SpikeToolkit Section). Taken together, the `SortingExtractor` is an object representation of a sorting output along with any associated post-processing.

Critically, both `Extractor` types can lazily query the underlying datasets for information as it is required, reducing their memory footprint and allowing their use for long, large-scale recordings. While this is the default operation mode, `Extractors` can also cache parts of the dataset in temporary binary files to enable faster downstream computations at the cost of higher memory usage. All extracted data is converted into either native Python data structures or into `numpy` arrays for immediate use in Python. Additionally, each `Extractor` can be dumped to and loaded from a `json` file, a `pickle` file, or a dictionary, ensuring full provenance and allowing for parallel processing.

The following code snippet illustrates how `Extractors` can be used to retrieve raw traces from an extracellular recording and spike trains from a sorting output:

```
import spikeinterface.extractors as se
recording = se.MyFormatRecordingExtractor(file_path='myrecording')
sorting = se.MyFormatSortingExtractor(file_path='mysorting')
traces = recording.get_traces() # 2D numpy array (channels x time)
spike_train = sorting.get_unit_spike_train(unit_id=1) # 1D numpy array
```

Along with using `Extractors` for single files, it is possible to access data from multiple files or portions of files with the `MultiExtractors` and `SubExtractors`, respectively. Both have identical functionality to normal `Extractors` and can be used and treated in the same ways, simplifying, for instance, the combined analysis of a recording split into multiple files.

As of this moment, SpikeInterface supports 19 extracellular recording formats and 18 sorting output formats. The available file formats can be found in *Table 1*. Although this covers many popular formats in extracellular analysis (including Neurodata Without Borders, *Teeters et al., 2015*, and *NIX, 2015*), we expect the number of formats to grow with future versions as adding a new format is as simple as making a new `Extractor` subclass for it. We also have started to integrate NEO's

**Table 1.** Currently available file formats in SpikeInterface and if they are writable.
*The Phy writing method is implemented in spiketoolkit as the export_to_phy function (all other writing methods are implemented in spikeextractors).

| Raw formats | Writable | Reference | Sorted formats | Writable | Reference |
|---|---|---|---|---|---|
| Klusta | Yes | *Rossant et al., 2016* | Klusta | Yes | *Rossant et al., 2016* |
| Mountainsort | Yes | *Jun et al., 2017a* | Mountainsort | Yes | *Jun et al., 2017a* |
| Phy* | Yes | *Rossant and Harris, 2013* | Phy* | Yes | *Rossant and Harris, 2013* |
| Kilosort/Kilosort2 | No | *Pachitariu et al., 2016*; *Rossant et al., 2014* | Kilosort/Kilosort2 | No | *Pachitariu et al., 2016*; *Rossant et al., 2014* |
| SpyKING Circus | No | *Yger et al., 2018* | SpyKING Circus | Yes | *Yger et al., 2018* |
| Exdir | Yes | *Dragly et al., 2018* | Exdir | Yes | *Dragly et al., 2018* |
| MEArec | Yes | *Buccino and Einevoll, 2020* | MEArec | Yes | *Buccino and Einevoll, 2020* |
| Open Ephys | No | *Siegle et al., 2017* | Open Ephys | No | *Siegle et al., 2017* |
| Neurodata Without Borders | Yes | *Teeters et al., 2015* | Neurodata Without Borders | Yes | *Teeters et al., 2015* |
| NIX | Yes | *NIX, 2015* | NIX | Yes | *NIX, 2015* |
| Plexon | No | *Plexon, 2020* | Plexon | No | *Plexon, 2020* |
| Neuralynx | No | *Neuralynx, 2020* | Neuralynx | No | *Neuralynx, 2020* |
| SHYBRID | Yes | *Wouters et al., 2020* | SHYBRID | Yes | *Wouters et al., 2020* |
| Neuroscope | Yes | *Hazan et al., 2006* | Neuroscope | Yes | *Hazan et al., 2006* |
| SpikeGLX | No | *Karsh, 2016* | HerdingSpikes2 | Yes | *Hilgen et al., 2017* |
| Intan | No | *Intan, 2010* | JRCLUST | No | *Jun et al., 2017b* |
| MCS H5 | No | *MCS, 2020* | Wave clus | No | *Chaure et al., 2018* |
| Biocam HDF5 | Yes | *Biocam, 2018* | Tridesclous | No | *Garcia and Pouzat, 2015* |
| MEA1k | Yes | *MEA1k, 2020* | NPZ (numpy zip) | Yes | N/A |
| MaxOne | No | *MaxWell, 2020* | | | |
| Binary | Yes | N/A | | | |

(*Garcia et al., 2014*) I/O system into `spikeextractors` which allow SpikeInterface to support many more open-source and proprietary file formats without changing any functionality. Already, two recording formats have been added through our NEO integration (*Neuralynx, 2020* and *Plexon, 2020*).

## SpikeToolkit
The `spiketoolkit` package (https://github.com/SpikeInterface/spiketoolkit; *Buccino et al., 2020b*) is designed for efficient pre-processing, post-processing, validation, and curation of extracellular datasets and sorting outputs. It contains four modules that encapsulate each of these functionalities: `preprocessing`, `postprocessing`, `validation`, and `curation`.

### Pre-processing
The `preprocessing` module provides functions to process raw extracellular recordings before spike sorting. To pre-process an extracellular recording, the user passes a `RecordingExtractor` to a pre-processing function which returns a new 'preprocessed' RecordingExtractor. This new `RecordingExtractor`, which can be used in exactly the same way as the original extractor, implements the preprocessing in a *lazy* fashion so that the actual computation is performed only when data is requested. As all pre-processing functions take in and return a `RecordingExtractor`, they can be naturally chained together to perform multiple pre-processing steps on the same recording.

Pre-processing functions range from commonly used operations, such as bandpass filtering, notch filtering, re-referencing signals, and removing channels, to more advanced procedures such as clipping traces depending on the amplitude, or removing artifacts arising, for example, from electrical

stimulation. The following code snippet illustrates how to chain together a few common pre-processing functions to process a raw extracellular recording:

```
import spikeinterface.spiketoolkit as st
recording = st.preprocessing.bandpass_filter(recording, freq_min=300, freq_max=6000)
recording_1 = st.preprocessing.remove_bad_channels(recording, bad_channels=[5])
recording_2 = st.preprocessing.common_reference(recording_1, reference='median')
```

## Post-processing

The `postprocessing` module provides functions to compute and store information about an extracellular recording given an associated sorting output. As such, post-processing functions are designed to take in both a `RecordingExtractor` and a `SortingExtractor`, using them in conjunction to compute the desired information. These functions include, but are not limited to: extracting unit waveforms and templates, computing principle component analysis projections, as well as calculating features from templates (e.g. peak to valley duration, full-width half maximum).

One essential feature of the `postprocessing` module is that it provides the functionality to export a `RecordingExtractor`/`SortingExtractor` pair into the `Phy` format for manual curation later. `Phy` (**Rossant and Harris, 2013**; **Rossant et al., 2016**) is a popular manual curation GUI that allows users to visualize a sorting output with several views and to curate the results by manually merging or splitting clusters. `Phy` is already supported by several spike sorters (including `klusta`, `Kilosort`, `Kilosort2`, and `SpyKING Circus`) so our exporter function extends `Phy`'s functionality to all SpikeInterface-supported spike sorters. After manual curation is performed in `Phy`, the curated data can be re-imported into SpikeInterface using the `PhySortingExtractor` for further analysis. The following code snippet illustrates how to retrieve waveforms for each sorted unit, compute principal component analysis (PCA) features for each spike, and export to `Phy` using SpikeInterface:

```
import spikeinterface.toolkit as st
waveforms = st.postprocessing.get_unit_waveforms(recording, sorting)
pca_scores = st.postprocessing.compute_unit_pca_scores(recording, sorting, n_comp=3)
st.postprocessing.export_to_phy(recording, sorting, output_folder='phy_folder')
```

## Validation

The `validation` module allows users to automatically evaluate spike sorting results in the absence of ground truth with a variety of quality metrics. The quality metrics currently available are a compilation of historical and modern approaches that were re-implemented by researchers at Allen Institute for Brain Science (https://github.com/AllenInstitute/ecephys_spike_sorting; **Siegle et al., 2019b**) and by the SpikeInterface team (see **Table 2**).

Each of SpikeInterface's quality metric functions internally utilize the `postprocessing` module to generate all data needed to compute the specified metric (amplitudes, principal components, etc.). The following code snippet demonstrates how to compute both a single quality metric (isolation distance) and also *all* the quality metrics with just two function calls:

```
import spikeinterface.toolkit as st
iso_metric = st.validation.compute_isolation_distances(sorting, recording)
all_metrics = st.validation.compute_quality_metrics(sorting, recording)
```

## Curation

The `curation` module allows users to quickly remove units from a `SortingExtractor` based on computed quality metrics. To curate a sorted dataset, the user passes a `SortingExtractor` to a curation function which returns a new 'curated' `SortingExtractor` (similar to how pre-processing works). This new `SortingExtractor` can be used in exactly the same way as the original extractor.

**Table 2.** Currently available quality metrics in Spikeinterface.
Re-implemented by researchers at Allen Institute for Brain and by the SpikeInterface team.

| Metric | Description | Reference |
|---|---|---|
| Signal-to-noise ratio | The signal-to-noise ratio computed on unit templates. | N/A |
| Firing rate | The average firing rate over a time period. | N/A |
| Presence ratio | The fraction of a time period in which spikes are present. | N/A |
| Amplitude Cutoff | An estimate of the miss rate based on an amplitude histogram. | N/A |
| Maximum drift | The maximum change in spike position (computed as the center of mass of the energy of the first principal component score) throughout a recording. | N/A |
| Cumulative drift | The cumulative change in spike position throughout a recording. | N/A |
| ISI violations | The rate of inter-spike-interval (ISI) refractory period violations. | *Hill et al., 2011* |
| Isolation Distance | Radius of the smallest ellipsoid that contains *all* the spikes from a cluster and an equal number of spikes from other clusters (centered on the specified cluster). | *Harris et al., 2001* |
| L-ratio | Assuming that the distribution of spike distances from a cluster center is multivariate normal, L-ratio is the average value of the tail distribution for non-member spikes of that cluster. | *Schmitzer-Torbert and Redish, 2004* |
| D-Prime | The classification accuracy between two units based on linear discriminant analysis (LDA) | *Hill et al., 2011* |
| Nearest-neighbors | A non-parametric estimate of unit contamination using nearest-neighbor classification. | *Chung et al., 2017* |
| Silhouette score | The ratio between cohesiveness of a cluster (distance between member spikes) and its separation from other clusters (distance to non-member spikes). | *Rousseeuw, 1987* |

As all curation functions take in and return a `SortingExtractor`, they can be naturally chained together to perform multiple curation steps on the same sorting output.

Currently, all implemented curation functions are based on excluding units with respect to a user-defined threshold on a specified quality metric. These curation functions will compute the associated quality metric and then threshold the dataset accordingly. The following code snippet demonstrates how to chain together two curation functions that are based on different quality metrics and apply a 'less' threshold to the underlying units (exclude all units below the given threshold):

```
import spikeinterface.toolkit as st
sorting_1 = st.curation.threshold_firing_rates(sorting, threshold=2.3, threshold_sign='less')
sorting_2 = st.curation.threshold_snrs(sorting_1, recording, threshold=10, threshold_sign='less')
```

## SpikeSorters

The `spikesorters` (https://github.com/SpikeInterface/spikesorters; *Buccino et al., 2020c*) package provides a straightforward interface for running spike sorting algorithms supported by SpikeInterface. Modern spike sorting algorithms are built and deployed in a variety of programming languages including C, C++, MATLAB, and Python. Along with variability in the underlying program languages, each sorting algorithm may depend on external technologies like CUDA or command line interfaces (CLIs), complicating standardization. To unify these disparate algorithms into a single codebase, `spikesorters` provides Python-wrappers for each supported spike sorting algorithm. These spike sorting wrappers use a standard API for running the corresponding algorithms, internally handling intrinsic complexities such as automatic code generation for MATLAB- and CLI-based algorithms. Each spike sorting wrapper is implemented as a subclass of a `BaseSorter` class that contains all shared code for running the spike sorters.

To run a specific spike sorting algorithm, users can pass a `RecordingExtractor` object to the associated function in `spikesorters` and overwrite any default parameters with new values (only essential parameters are exposed to the user for modification). Internally, each function initializes a spike sorting wrapper with the user-defined parameters. This wrapper then creates and modifies a

new spike sorter configuration and runs the sorter on the dataset encapsulated by the `Recordin-gExtractor`. Once the spike sorting algorithm is finished, the sorting output is saved and a corresponding `SortingExtractor` is returned to the user. For each sorter, all available parameters and their descriptions can be retrieved using the `get_default_params()` and `get_params_description()` functions, respectively.

In the following code snippet, Mountainsort4 and Kilosort2 are used to sort an extracellular recording. Running each algorithm (and changing the default parameters) can be done as follows:

```
import spikeinterface.sorters as ss
sorting_MS4 = ss.run_mountainsort4(recording, adjacency_radius=50)
sorting_KS2 = ss.run_kilosort2(recording, detect_threshold=5)
```

Our spike sorting functions also allow for users to sort specific 'groups' of channels in the recording separately (and in parallel, if specified). This can be very useful for multiple tetrode recordings where the data are all stored in one file, but the user wants to sort each tetrode separately. For large-scale analyses where the user wants to run many different spike sorters on many different datasets, `spikesorters` provides a launcher function which handles any internal complications associated with running multiple sorters and returns a nested dictionary of `SortingExtractor` objects corresponding to each sorting output. The launcher can be deployed on HPC platforms through the `multiprocessing` or `dask` engine (*Dask, 2016*). Finally, and importantly, when running a spike sorting job the recording information and all the spike sorting parameters are saved in a log file, including the console output of the spike sorting run (which can be used to inspect errors). This provenance mechanism ensures full reproducibility of the spike sorting pipeline.

Currently, SpikeInterface supports 10 semi-automated spike sorters which are listed in *Table 3*. We encourage developers to contribute to this expanding list in future versions and we provide comprehensive documentation on how to do so (https://spikeinterface.readthedocs.io/en/latest/contribute.html).

## SpikeComparison

The `spikecomparison` package (https://github.com/SpikeInterface/spikecomparison; *Buccino et al., 2020d*) provides a variety of tools that allow users to compare and benchmark sorting outputs. Along with these comparison tools, `spikecomparison` also provides the functionality

**Table 3.** Currently available spike sorters in Spikeinterface.
TM = Template Matching; SL = Spike Localization; DB = Density-based clustering.

| Name | Method | Notes | Reference |
|---|---|---|---|
| Klusta | DB | Python-based, semi-automatic, designed for low channel count, dense probes. | *Rossant et al., 2016* |
| Mountainsort4 | DB | Python-based, fully automatic, unique clustering method (isosplit), designed for low channel count, dense probes and tetrodes. | *Chung et al., 2017* |
| Kilosort | TM | MATLAB-based, GPU support, semi-automated final curation. | *Pachitariu et al., 2016* |
| Kilosort2 | TM | MATLAB-based, GPU support, semi-automated final curation, designed to correct for drift. | *Pachitariu et al., 2018* |
| SpyKING Circus | TM | Python-based, fast and scalable with CPUs, designed to correct for drift. | *Yger et al., 2018* |
| HerdingSpikes2 | DB + SL | Python-based, fast and scalable with CPUs, scales up to thousands of channels. | *Hilgen et al., 2017* |
| Tridesclous | TM | Python-based, graphical user interface, GPU support, multi-platform | *Garcia and Pouzat, 2015* |
| IronClust | DB + SL | MATLAB-based, GPU support, designed to correct for drift. | *Jun et al., 2020* |
| Wave clus | TM | Matlab-based, fully automatic, designed for single electrodes and tetrodes, multi-platform. | *Chaure et al., 2018* |
| HDsort | TM | Matlab-based, fast and scalable, designed for large-scale, dense arrays. | *Diggelmann et al., 2018* |

to run systematic performance comparisons of multiple spike sorters on multiple ground-truth recordings.

Within spikecomparison, there exist three core comparison functions:

1. `compare_two_sorters` - Compares two spike sorting outputs.
2. `compare_multiple_sorters` - Compares multiple spike sorting outputs.
3. `compare_sorter_with_ground_truth` - Compares a spike sorting output to ground truth.

Each of these comparison functions takes in multiple `SortingExtractor` objects and uses them to compute agreement scores among the underlying spike trains. The agreement score between two spike trains is defined as:

$$score = \frac{\#n_{matches}}{\#n_1 + \#n_2 - \#n_{matches}} \tag{1}$$

where $\#n_{matches}$ is the number of 'matched' spikes between the two spike trains and $\#n_1$ and $\#n_2$ are the number of spikes in the first and second spike train, respectively. Two spikes from two different spike trains are 'matched' when they occur within a certain time window of each other (this window length can be adjusted by the user and is 0.4 ms by default).

When comparing two sorting outputs (`compare_two_sorters`), a linear assignment based on the Hungarian method (*Kuhn, 1955*) is used. With this assignment method, each unit from the first sorting output can be matched to at most one other unit in the second sorting output. The final result of this comparison is then the list of matching units (given by the Hungarian method) and the agreement scores of the spike trains.

The multi-sorting comparison function (`compare_multiple_sorters`) can be used to compute the agreement among the units of many sorting outputs at once. Internally, pair-wise sorter comparisons are run for all of the sorting output pairs. A graph is then built with the sorted units as nodes and the agreement scores among the sorted units as edges. With this graph implementation, it is straightforward to query for units that are in agreement among multiple sorters. For example, if three sorting outputs are being compared, any units that are in agreement among all three sorters will be part of a subgraph with large weights.

For a ground-truth comparison (`compare_sorter_with_ground_truth`), either the Hungarian or the best-match method can be used. With the Hungarian method, each tested unit from the sorting output is matched to at most a single ground-truth unit. With the best-match method, a tested unit from the sorting output can be matched to multiple ground-truth units (above an adjustable agreement threshold) allowing for more in-depth characterizations of sorting failures. Note that in the SpikeForest benchmarking software suite (*Magland et al., 2020*), the best-match strategy is used.

Additionally, when comparing a sorting output to a ground-truth sorted result, each spike can be optionally labeled as:

- True positive (*tp*): Found both in the ground-truth spike train and tested spike train.
- False negative (*fn*): Found in the ground-truth spike train, but not in the tested spike train.
- False positive (*fp*): Found in the tested spike train, but not in the ground-truth spike train.

Using these labels, the following performance measures can be computed:

- Accuracy: $\frac{\#tp}{(\#tp+\#fn+\#fp)}$
- Recall: $\frac{\#tp}{(\#tp+\#fn)}$
- Precision: $\frac{\#tp}{(\#tp+\#fp)}$
- Miss rate: $\frac{\#fn}{(\#tp+\#fn)}$
- False discovery rate: $\frac{\#fp}{(\#tp+\#fp)}$

While previous metrics give a measure of individual spike train quality, we also propose metrics at a unit population level. Based on the matching results and the scores, the units of the sorting output are classified as *well-detected*, *false positive*, *redundant*, and *overmerged*. Well-detected units are matched units with an agreement score above 0.8. False positive units are unmatched units or units which are matched with an agreement score below 0.2. Redundant units have agreement scores above 0.2 with only one ground-truth unit, but are not the best matched tested units (redundant

units can either be oversplit or duplicate units). Overmerged units have an agreement score above 0.2 with two or more ground-truth units. All these agreement score thresholds are adjustable by the user. We highlight to the reader that the unit classification proposed here is currently only based on agreement score (i.e. accuracy). More sophisticated classification rules could involve a combination of accuracy, precision, and recall values, which can be easily computed for each unit with the `spikecomparison` module.

The following code snippet shows how to perform all three types of spike sorter comparisons:

```
import spikeinterface.comparison as sc
comp_type_1 = sc.compare_two_sorters(sorting1, sorting2)
comp_type_2 = sc.compare_multiple_sorters([sorting1, sorting2, sorting3])
comp_type_3 = sc.compare_sorter_with_ground_truth(gt_sorting, tested_sorting)
```

Along with the three comparison functions, `spikecomparison` also includes a `GroundTruthStudy` class that allows for the systematic comparison of multiple spike sorters on multiple ground-truth datasets. With this class, users can set up a study folder (in which the recordings to be tested are saved), run several spike sorters and store their results in a compact way, perform systematic ground-truth comparisons, and aggregate the results in pandas dataframes (*McKinney, 2010*).

### SpikeWidgets

The `spikewidgets` package (https://github.com/SpikeInterface/spikewidgets; *Buccino et al., 2020e*) implements a variety of widgets that allow for efficient visualization of different elements in a spike sorting pipeline.

There exist four categories of widgets in `spikewidgets`. The first category utilizes a `RecordingExtractor` for its visualization. This category includes widgets for visualizing time series data, electrode geometries, signal spectra, and spectrograms. The second category utilizes a `SortingExtractor` for its visualization. These widgets include displays for raster plots, auto-correlograms, cross-correlograms, and inter-spike-interval distributions. The third category utilizes both a `RecordingExtractor` and a `SortingExtractor` for its visualization. These widgets include visualizations of unit waveforms, amplitude distributions for each unit, amplitudes of each unit over time, and PCA features. The fourth category utlizes comparison objects from the `spikecomparison` package for its visualization. These widgets allow the user to visualize confusion matrices, agreement scores, spike sorting performance metrics (e.g. accuracy, precision, recall) with respect to a unit property (e.g. SNR), and the agreement between multiple sorting algorithms on the same dataset.

The following code snippet demonstrates how SpikeInterface can be used to visualize ten seconds of both the extracellular traces and the corresponding raster plot:

```
import spikeinterface.widgets as sw
sw.plot_timeseries(recording, channel_ids=[0,1,2,3], trange=[0,10])
sw.plot_rasters(sorting, unit_ids=[0,1,3], trange=[0,10]).
```

## Building a spike sorting pipeline

So far, we have given an overview of each of the main packages in isolation. In this section, we illustrate how these packages can be combined, using both the Python API and the `Spikely` GUI, to build a robust spike sorting pipeline. The spike sorting pipeline that we construct using SpikeInterface is depicted in *Figure 6A* and consists of the following analysis steps:

1. Loading an Open Ephys recording (*Siegle et al., 2017*).
2. Loading a probe file.
3. Applying a bandpass filter.
4. Applying common median referencing to reduce the common mode noise.
5. Spike sorting with `Mountainsort4`.
6. Removing clusters with less than 100 events.
7. Exporting the results to Phy for manual curation.

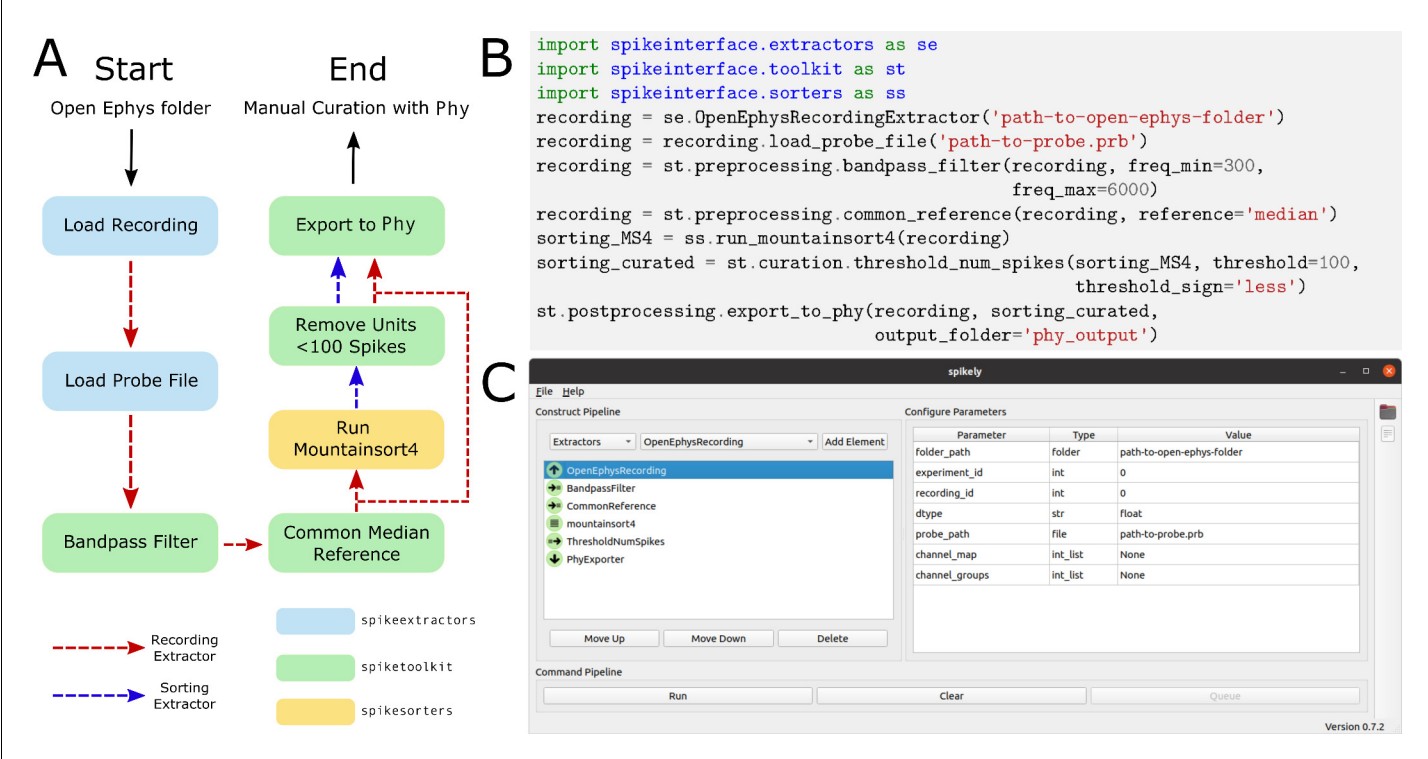

**Figure 6.** Sample spike sorting pipeline using SpikeInterface. (**A**) A diagram of a sample spike sorting pipeline. Each processing step is colored to represent the SpikeInterface package in which it is implemented and the dashed, colored arrows demonstrate how the Extractors are used in each processing step. (**B**) How to use the Python API to build the pipeline shown in (**A**). (**C**) How to use the GUI to build the pipeline shown in (**A**).

Traditionally, implementing this pipeline is challenging as the user has to load data from multiple file formats, interface with a probe file, memory-map all the processing functions, prepare the correct inputs for Mountainsort4, and understand how to export the results into Phy. Even if the user manages to implement all of the analysis steps on their own, it is difficult to verify their correctness or reuse them without proper unit testing and code reviewing.

## Using the Python API

Using SpikeInterface's Python API to build the pipeline shown in *Figure 6A* is straightforward. Each of the seven steps is implemented with a single line of code (as shown in *Figure 6B*). Additionally, data visualizations can be added for each step of the pipeline using the appropriate widgets (as described in the SpikeWidgets Section). Unlike handmade scripts, SpikeInterface has a wide range of unit tests, employs continuous integration, and has been carefully developed by a team of researchers. Users, therefore, can have increased confidence that the pipelines they create are correct and reusable. Additionally, SpikeInterface tracks the entire provenance of the performed analysis, allowing other users (or the same user) to reproduce the analysis at a later date.

## Using the spikely GUI

Along with our Python API, we also developed `spikely` (https://github.com/SpikeInterface/spikely; *Hurwitz et al., 2020*), a PyQt-based GUI that allows for simple construction of complex spike sorting pipelines. With `spikely`, users can build workflows that include: (i) loading a recording and a probe file; (ii) performing pre-processing on the underlying recording with multiple processing steps; (iii) running any spike sorter supported by SpikeInterface on the processed recording; (iv) automatically curating the sorter's output; and (v) exporting the final result to a variety of file formats, including Phy. At its core, `spikely` utilizes SpikeInterface's Python API to run any constructed spike sorting workflow. This ensures that the functionality of spikely grows organically with that of SpikeInterface.

*Figure 6C* shows a screenshot from `spikely` where the pipeline in *Figure 6A* is constructed. Each stage of the pipeline is added using drop-down lists, and all the parameters (which were not left at their default values) are set in the right-hand panel. Once a pipeline is constructed in `spikely`, the user can save it using the built-in save functionality and then load it back into spikely at a later date. Since `spikely` is cross-platform and user-friendly, we believe it can be utilized to increase the accessibility and reproducibility of spike sorting.

## Discussion

In this paper, we introduced SpikeInterface, a Python framework designed to enhance the accessibility, reliability, efficiency, and reproducibility of spike sorting. To illustrate the use-cases and advantages of SpikeInterface, we performed a detailed meta-analysis that included: quantifying the agreement among six modern sorters on a real dataset, benchmarking each sorter on a simulated ground-truth recording, and investigating the performance of a consensus-based spike sorting and how it compares with manually curated results. To highlight the modular design of SpikeInterface, we then provided descriptions and code samples for each of the five main packages and showed how they could be chained together to construct flexible spike sorting workflows.

### Ensemble spike sorting

Our analysis demonstrated that spike sorters not only differ in unit isolation quality, but can also return a significant number of false positive units. To identify true neurons and remove poorly sorted and noisy units, we combined the output of several spike sorters and found that although agreement between sorters is generally poor, units that are found by more than one sorter are likely positives. This strategy, which we term consensus-based or ensemble spike sorting (a terminology borrowed from machine learning; *Dietterich, 2000*) appears to be a viable alternative to manual curation which suffers from high-variability among different operators (*Wood et al., 2004*; *Rossant et al., 2016*). Alternatives to manual curation are especially enticing as the density and number of simultaneously recording channels continue to increase rapidly.

We propose that consensus-based spike sorting (or curation) can be utilized in a number of different ways. A first possibility is to choose a suitable spike sorter (for instance, based on the extensive ground-truth comparison performed by SpikeForest; *Magland et al., 2020*) and then to curate its output by retaining the units that are in agreement with other sorters. Alternatively, a more conservative approach is to simply record the agreement scores for all sorted units and then *hand-curate* only those units that have low agreement. A third method, already implemented in SpikeInterface, is to generate a consensus spike sorting by using, for each unit, the union of the two closest matching units from different sorters (matching spikes are only considered once). Although more work is needed to quantitatively assess the advantages and disadvantages of each approach, our analysis indicates that agreement among sorters can be a useful tool for curating sorting results.

Although ensemble spike sorting is an exciting new direction to explore, there are other methods for curation that must be considered. One popular curation method is to accept or reject sorted units based on a variety of quality metrics (this is supported by SpikeInterface). Another method that is gaining more popularity is to use the large amount of available curated datasets to train classifiers that can automatically flag a unit as 'good' or 'noise' depending on some features, such as waveform shape. Finally, while manual curation is subjective and time consuming, it is the only method that allows for merging and splitting of units and, through powerful software tools such as Phy (*Rossant et al., 2014*; *Rossant et al., 2016*), it allows for full control over the curation process. Future research into these different curation methods is required to determine which are appropriate for the new influx of high-density extracellular recording devices.

### Comparison to other frameworks

As mentioned in the introduction, many software tools have attempted to improve the accessibility and reproducibility of spike sorting. Here, we review the four most recent tools that are in use (to our knowledge) and compare them to SpikeInterface.

`Nev2lkit` (*Bongard et al., 2014*) is a cross-platform, C++-based GUI designed for the analysis of recordings from multi-shank multi-electrode arrays (Utah arrays). In this GUI, the spike sorting step consists of PCA for dimensionality reduction and then `klustakwik` for automatic clustering

(*Rossant et al., 2016*). As `Nev2lkit` targets low-density probes where each channel is spike sorted separately, it is not suitable for the analysis of high-density recordings. Also, since it implements only one spike sorter, users cannot utilize any consensus-based curation or exploration of the data. The software is available online (http://nev2lkit.sourceforge.net/), but it lacks version-control and automated testing with continuous integration platforms.

`SigMate` (*Mahmud et al., 2012*) is a MATLAB-based toolkit built for the analysis of electrophysiological data. SigMate has a large scope of usage including the analysis of electroencephalograpy (EEG) signals, local field potentials (LFP), and spike trains. Despite its broad scope, or because of it, the spike sorting step in `SigMate` is limited to `Wave` clus *Chaure et al., 2018*, which is mainly designed for spike sorting recordings from a few channels. This means that both major limitations of `Nev2lkit` (as discussed above) also apply to `SigMate`. The software is available online (https://sites.google.com/site/muftimahmud/codes), but again, it lacks version-control and automated testing with continuous integration platforms.

*Regalia et al., 2016* developed a spike sorting framework with an intuitive MATLAB-based GUI. The spike sorting functionality implemented in this framework includes four feature extraction methods, three clustering methods, and one template matching classifier (`O-Sort`; *Rutishauser et al., 2006*). These 'building blocks' can be combined to construct new spike sorting pipelines. As this framework targets low-density probes where signals from separate electrodes are spike sorted separately, its usefulness for newly developed high-density recording technology is limited. Moreover, this framework only runs with a specific file format (MCD format from Multi Channel Systems; *MCS, 2020*). The software is distributed upon request.

Most recently, *Nasiotis et al., 2019a* implemented `IN-Brainstorm`, a MATLAB-based GUI designed for the analysis of invasive neurophysiology data. `IN-Brainstorm` allows users to run three spike sorting packages (`Wave` clus [*Chaure et al., 2018*], UltraMegaSort2000 [*Hill et al., 2011*], and `Kilosort` [*Pachitariu et al., 2016*]). Recordings can be loaded and analyzed from six different file formats: Blackrock, Ripple, Plexon, Intan, NWB, and Tucker Davis Technologies. `IN-Brainstorm` is available on GitHub (https://github.com/brainstorm-tools/brainstorm3; *Nasiotis et al., 2019b*) and its functionality is documented (https://neuroimage.usc.edu/brainstorm/e-phys/Introduction). `IN-Brainstorm` does not include the latest spike sorting software (*Rossant et al., 2016*; *Yger et al., 2018*; *Chung et al., 2017*; *Jun et al., 2017b*; *Pachitariu et al., 2018*; *Hilgen et al., 2017*) (`IN-Brainstorm` does include instructions on how to import data that has been spike sorted by a non-supported spike sorter), and it does not support any post-sorting analysis such as quality metric calculation, automated curation, or sorting output comparison.

## Outlook

As it stands, spike sorting is still an open problem. No step in the spike sorting pipeline is completely solved and no spike sorter can be used for all applications. With SpikeInterface, researchers can quickly build, run, and evaluate many different spike sorting workflows on their specific datasets and applications, allowing them to determine which will work best for them. Once a researcher determines an ideal workflow for their specific problem, it is straightforward to share and re-use that workflow in other laboratories as the full provenance is automatically stored by SpikeInterface. We envision that many laboratories will use SpikeInterface to satisfy their spike sorting needs.

Along with its applications to extracellular analysis, SpikeInterface is also a powerful tool for developers looking to create new spike sorting algorithms and analysis tools. Developers can test their methods using our efficient and comprehensive comparison functions. Once satisfied with their performance, developers can integrate their work into SpikeInterface, allowing them access to a large-community of new users and providing them with automatic file I/O for many popular extracellular dataset formats. For developers who work on projects that utilize spike sorting, SpikeInterface is useful out-of-the-box, providing more reliability and functionality than lab-specific scripts. We envision that many developers will be excited to use and integrate with SpikeInterface.

Already, SpikeInterface is being used in a variety of applications. The file IO, preprocessing, and spike sorting capabilities of SpikeInterface are an integral part of SpikeForest (*Magland et al., 2020*), which is an interactive website for benchmarking and tracking the accuracy of publicly available spike sorting algorithms. At present, this project includes ten spike sorting algorithms and more than 300 extracellular recordings with ground-truth firing information. SpikeInterface's ability to read and write to a multitude of extracellular file formats is also being utilized by Neurodata

Without Borders (*Teeters et al., 2015*) in their `nwb-conversion-tools` package. We hope to continue integrating SpikeInterface into cutting-edge extracellular analysis frameworks.

## Acknowledgements

This work was supported by the Wellcome Trust grant 214431/Z/18/Z (MHH). APB is supported by an ETH Zurich Postdoctoral Fellowship 19–2 FEL-17, and by the Simula-UCSD-University of Oslo Research and PhD training (SUURPh) program, funded by the Norwegian Ministry of Education and Research. CLH is supported by the Thouron Award and by the Institute for Adaptive and Neural Computation, University of Edinburgh. JHS wishes to thank the Allen Institute founder, Paul G Allen, for his vision, encouragement and support. We thank Shangmin Guo for his recent contributions to debugging and improving the codebase.

## Additional information

### Funding

| Funder | Grant reference number | Author |
| --- | --- | --- |
| Wellcome Trust | 214431/Z/18/Z | Matthias H Hennig |
| ETH Zürich | 19–2 FEL-17 | Alessio P Buccino |
| University of Oslo | PhD training (SUURPh) program | Alessio P Buccino |
| Norwegian Ministry of Education, Research and Church Affairs | | Alessio P Buccino |
| University of Edinburgh | Thouron Award | Cole L Hurwitz |

The funders had no role in study design, data collection and interpretation, or the decision to submit the work for publication.

### Author contributions

Alessio P Buccino, Conceptualization, Resources, Data curation, Software, Visualization, Methodology, Writing - original draft, Writing - review and editing; Cole L Hurwitz, Conceptualization, Resources, Software, Visualization, Methodology, Writing - original draft, Writing - review and editing; Samuel Garcia, Software, Visualization, Methodology, Writing - review and editing; Jeremy Magland, Conceptualization, Software, Methodology, Writing - review and editing; Joshua H Siegle, Data curation, Software, Methodology, Writing - review and editing; Roger Hurwitz, Software; Matthias H Hennig, Conceptualization, Resources, Software, Supervision, Visualization, Writing - original draft, Writing - review and editing

### Author ORCIDs

Alessio P Buccino https://orcid.org/0000-0003-3661-527X
Cole L Hurwitz https://orcid.org/0000-0002-2023-1653
Samuel Garcia https://orcid.org/0000-0001-6389-9779
Jeremy Magland http://orcid.org/0000-0002-5286-4375
Joshua H Siegle https://orcid.org/0000-0002-7736-4844
Matthias H Hennig https://orcid.org/0000-0001-7270-5817

### Decision letter and Author response

Decision letter https://doi.org/10.7554/eLife.61834.sa1
Author response https://doi.org/10.7554/eLife.61834.sa2

## Additional files

### Supplementary files

• Transparent reporting form

### Data availability

All data generated or analysed during this study are included in the manuscript and supporting files. The datasets are uploaded to the DANDI archive, dataset 000034 (https://gui.dandiarchive.org/#/dandiset/000034). The source code for generating all figures is also publicly available at: https://spikeinterface.github.io/.

The following dataset was generated:

| Author(s) | Year | Dataset title | Dataset URL | Database and Identifier |
|---|---|---|---|---|
| Buccino AP, Hurwitz CL, Garcia S, Magland J, Siegle JH, Hurwitz R, Hennig MH | 2020 | SpikeInterface, a unified framework for spike sorting | https://gui.dandiarchive.org/#/dandiset/000034 | DANDI, 000034 |

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
