## [Decision Letter]

**Acceptance summary:**

SpikeInterface is an integrated set of tools that makes it straightforward for researchers to set up a complete spike sorting workflow. SpikeInterface supports many common data formats and modern spike sorters and provides post-processing tools for characterization of the spike sorting results. This allows for validation and comparison of multiple spike sorting results. Results suggest that combining the results of multiple spike sorters could help to reduce the number of false positive units, which is an interesting future direction that will likely inspire further investigation. This tool is expected to be useful for researchers in many different areas who are studying neuronal responses.

**Decision letter after peer review:**

[Editors’ note: the authors submitted for reconsideration following the decision after peer review. What follows is the decision letter after the first round of review.]

Thank you for submitting your work entitled "SpikeInterface, a unified framework for spike sorting" for consideration by *eLife*. Your article has been reviewed by a Senior Editor, a Reviewing Editor, and three reviewers. The following individuals involved in review of your submission have agreed to reveal their identity: Fabian Kloosterman (Reviewer #2).

Our decision has been reached after consultation between the reviewers. Based on these discussions and the individual reviews below, we regret to inform you that your work will not be considered further for publication in *eLife*.

There was a great deal of discussion about this manuscript among the reviewers and the editor after the individual reviews were received. Ultimately, the consensus was that this work in its present form is too preliminary to be useful to, and to make a major impact on, a broad range of users. At *eLife*, the standard revision period is approximately two months, and therefore papers are largely assessed "as is" to allow authors to decide when to publish the work at the stage when they feel it is ready. In this case, though, reviewers agreed that the work needs a number of major revisions that constitute a substantial amount of work in order to make a major impact across a broad range of readers (e.g., reviewers were not confident that this tool is ready to be used by anyone who is recording with Neuropixels). If you agree with the reviewers that major changes to the tool are necessary to make a major impact on the field, then we would encourage you to submit a majorly revised manuscript to us in the future, citing this manuscript number and requesting the same editor. We would be willing to re-assess the manuscript at that time. Otherwise, you can just move on to a more specialized journal, keeping your tool in its present form and perhaps improving on it in future publications.

The three original reviews are included in their entirety below. However, due to the extensive and constructive discussion that occurred after reviewers read each other's reviews, we would like to emphasize a number of interrelated major points that were discussed in the consultation:

1) A concern was raised that SpikeInterface limits the flexibility of the spike sorters it contains and makes spike sorting more of a "black box". Given the lack of real ground truth available for results comparison, this was viewed as a major weakness. Reviewers felt that users still need to be able to look carefully at the units, understand the different algorithms, and properly set the parameters. Using the default parameters could lead to suboptimal results, and the authors did not attempt to adjust parameters. Reviewers felt that the SpikeInterface toolbox could also be used to compare results of the same spike sorter using different parameters and that this would be useful to find optimal parameters and would increase the potential impact of this tool.

2) The analyses and presentation of the comparison of spike sorters was viewed as weak. Reviewers agreed that careful manual curation is still the only right way to compare spike sorting results. Reviewers felt that it would be a mistake for readers who are new to the spike sorting process to look at the analyses shown in the paper as the right way to compare spike sorting results. Reviewers felt that a manual curation step is necessary to make the spike sorter evaluation process more useful.

3) It was suggested that the authors should attempt to perform some sort of smart cluster merging strategy that utilizes the output of different sorters.

4) Another suggestion for a potentially useful addition was that users would be able to swap in and out different algorithmic components in the spike sorting pipeline. Combined with manual curation and comparison to "ground truth" data sets, reviewers felt that this could help users to determine the best algorithmic components for particular types of recordings.

Reviewer #1:

In this paper, the authors introduce a Python package for easily running many different spike sorters and exporting to many different formats. The goal is to make it easier for electrophysiologists to run their data through the spike sorters and output these results to Phy and other GUIs for data visualizations. While I agree that spike sorting is a hard problem and users need to be helped as much as possible, I don't think this framework helps a lot and will ultimately not find much use. I think Phy (already published and widely used) does most of the work that the authors suggest SpikeInterface should do, and in fact the main use case for SpikeInterface seems to be as an exporter to Phy. At its core, the code provided here is a set of file converters and code wrappers that further obfuscate the black-boxes that many spike sorters are, and make it more difficult for users to know how to build a successful spike sorting pipeline for their own data.

Reviewer #2:

The work presented in this manuscript is of great interest to both spike sorting users and developers. The unified framework bridges the gap between the plethora of recording file formats and spike sorting packages, which is a major improvement in terms of spike sorting experience. The framework also features many interesting features related to spike sorting for processing recordings and sorting results. The manuscript is clearly written and introduces the functionality that is at hand in the framework in a concise way. Below is a list of major and minor comments that need to be addressed, however.

1) Spikeinterface is portrayed as a general spike sorting framework. Still, the spike sorting workflow supported by spikeinterface appears to be geared towards specific kind of data and sorters, i.e. those that work on high electrode count continuous datasets. The authors should make explicit the assumptions that are made in spikeinterface regarding the data that is accepted (e.g. datasets with only waveform snippets appear not to be supported) and the minimal requirement for spike sorters (e.g. do spike sorters need to include their own spike detection algorithm and spike feature extraction?).

2) The authors have chosen to run the spike sorters with their default parameters and without manual or automated refinement (i.e. noise cluster rejection, cluster merging/splitting). As many spike sorting algorithms explicitly depend on a manual cluster merging/splitting step after they have been applied to the data, it would be interesting to also provide an automated cluster merging (e.g., based on the ground truth as in Wouters, Kloosterman and Bertrand, 2019). This will improve the understanding of the true potential of a spike sorting algorithm, when comparing it to others in a ground-truth study. As a bare minimum, the authors should discuss the need of a post-sorting split/merge curation step and discuss the effect of leaving the step out on their results. Without such discussion, it would be premature to talk about a "consensus-based strategy" to select clusters (subsection “Application 1: Comparing Spike Sorters on Neuropixels Data”).

3) The authors define an agreement score to match clusters from different sorters and use the score to classify clusters (as compared to ground truth) as "well-detected", "false positive", "redundant" and "over-merged". However, a low agreement score could result from a high number of false positive detections or a high number of false negative detections (or both), and the interpretation would be different in these cases. In the extremes of no false positives or false negatives, an agreement score of 0.2 could either mean all spikes in a cluster represent 20% of the ground truth spikes (i.e. a clean partial cluster) or it could mean that all ground truth spikes represent 20% of the spikes in a cluster (i.e. a "dirty" over-merged cluster). Thus, the agreement score is not a good metric for the classification of the clusters. Instead, the authors should consider a classification based on different metric(s), e.g. both precision and recall.

4) We do not find the swarm plot in Figure 4 that compares the accuracy, precision and recall for multiple sorters very informative. First, the number of non-matched clusters is not obvious in this plot (we assume point with zero score are non-matched?). More importantly, there is often a trade-off between the number of false positive and false negatives, and each sorter may make a different trade-off, depending on the parameters. The swarm plot does not show the relation between precision and recall for each sorter, and a precision-recall scatter plot would be more informative.

Reviewer #3:

This submission describes a software toolbox aimed to facilitate the comparison of spike sorting algorithms. It is targeted for a broad user base, who may not have the time or technical ability to make such comparisons on their own. This tool addresses a need of the neuroscience community. Outlined below are number of suggested corrections.

Introduction: Not all the listed sorters are truly fully-manual, ie Mclust is semiautomatic.

Subsection “Overview of SpikeInterface”: Roman numerals swapped for spikecomparison and spikewidgets.

Subsection “SpikeExtractors”: It is unclear how recordingextractor, a visualization tool, provides functionality required to excess data to evaluate the spike sorting pipeling. This becomes more clear later, but could be made more clear sooner.

Subsection “SpikeExtractors” and subsection “SpikeToolkit”: The code snippets could be expanded to give more context and be more relevant.

Subsection “Curation”: Instead of holding of for the future, this functionality would be nice to implement here, if it is not an unreasonable amount of work.

Subsection “Using the Python API”: It could be said that spikeinterface is also handmade, maybe clarify the point.

Figure 3B is hard to read.

Figure 3D, what are the color code agreement levels exactly, this is unclear.

In Figure 4 it would be nice to see plotted SNR vs agreement score.

[Editors’ note: further revisions were suggested prior to acceptance, as described below.]

Thank you for submitting your article "SpikeInterface, a unified framework for spike sorting" for consideration by *eLife*. Your article has been reviewed by Laura Colgin as the Senior Editor and Reviewing Editor and three reviewers. The following individuals involved in review of your submission have agreed to reveal their identity: Sonja Grün (Reviewer #2); Fabian Kloosterman (Reviewer #3).

The reviewers have discussed the reviews with one another and the Reviewing Editor has drafted this decision to help you prepare a revised submission.

Summary:

The authors describe SpikeInterface, which is an integrated set of tools that makes it straightforward for researchers to set up a complete spike sorting workflow. SpikeInterface is modular (and extendible) and supports many common data formats and modern spike sorters. It provides postprocessing tools for characterization of the spike sorting results and for validation and comparison of multiple spike sorting results (e.g. against ground truth). SpikeInterface allows users to focus on the spike sorting results and curation, rather than having to glue together or (re)implement disparate tools themselves.

Compared to the previous version, the authors have now more clearly outlined the goals of SpikeInterface in the introduction. In addition to comparing multiple sorters, they have added new results that indicate that combining the results of multiple spike sorters ("ensemble spike sorting") could help to reduce the number of false positive units, which is an interesting future direction that needs further investigation.

Essential revisions:

1) Regarding the ensemble spike sorting approach:

– From the results shown in Figure 3C, it seems that one won't need to run all 6 sorters to eliminate the false positives. Could the authors quantify the relative benefit of combining 2, 3 or more sorters over the use of a single sorter?

– One could imagine that combining the results of spike sorters that use a different class of algorithm would provide more benefit than combining two sorters that use the same algorithm. Do the authors observe this?

– If reviewers understand correctly, many spike sorters will return a slightly different output when run a second time on the same data set. To what extent does running the same sorter twice give the same benefit (i.e. low agreement on false positive clusters) as running two different sorters?

– Would using the ensemble spike sorting approach give similar results as using a more stringent selection of units found by a single sorter (e.g. based on cluster quality metrics, SNR, spike amplitude, etc.)?

2) The authors used an arbitrary agreement score threshold of 0.5, which they acknowledge is a pragmatic but not necessarily the best choice to match units found by multiple sorters. Reviewers did not think it is necessary to change the way units are matched, but to provide more insight into the matching process it would be helpful to know what the distribution of unit agreement scores for matching pairs looks like.

3) When evaluating the spike sorting results on the simulated data set, the authors only mention matches and false positives. Reviewers did not see a mention of the number of false negatives (i.e. number of ground units that were missed by the sorters). Could the authors also indicate to what extent spikes of the missed clusters actually show up as part of the false positive units (e.g. because false positive units are actually overly split/merged true units)?

---

## [Author Response]

[Editors’ note: the authors resubmitted a revised version of the paper for consideration. What follows is the authors’ response to the first round of review.]

There was a great deal of discussion about this manuscript among the reviewers and the editor after the individual reviews were received. Ultimately, the consensus was that this work in its present form is too preliminary to be useful to, and to make a major impact on, a broad range of users. At eLife, the standard revision period is approximately two months, and therefore papers are largely assessed "as is" to allow authors to decide when to publish the work at the stage when they feel it is ready. In this case, though, reviewers agreed that the work needs a number of major revisions that constitute a substantial amount of work in order to make a major impact across a broad range of readers (e.g., reviewers were not confident that this tool is ready to be used by anyone who is recording with Neuropixels). If you agree with the reviewers that major changes to the tool are necessary to make a major impact on the field, then we would encourage you to submit a majorly revised manuscript to us in the future, citing this manuscript number and requesting the same editor. We would be willing to re-assess the manuscript at that time. Otherwise, you can just move on to a more specialized journal, keeping your tool in its present form and perhaps improving on it in future publications.The three original reviews are included in their entirety below. However, due to the extensive and constructive discussion that occurred after reviewers read each other's reviews, we would like to emphasize a number of interrelated major points that were discussed in the consultation:1) A concern was raised that SpikeInterface limits the flexibility of the spike sorters it contains and makes spike sorting more of a "black box". Given the lack of real ground truth available for results comparison, this was viewed as a major weakness. Reviewers felt that users still need to be able to look carefully at the units, understand the different algorithms, and properly set the parameters. Using the default parameters could lead to suboptimal results, and the authors did not attempt to adjust parameters. Reviewers felt that the SpikeInterface toolbox could also be used to compare results of the same spike sorter using different parameters and that this would be useful to find optimal parameters and would increase the potential impact of this tool.

We disagree that SpikeInterface limits the flexibility of the underlying sorter in any meaningful way. A key feature of SpikeInterface is that the Python-based wrappers for each spike sorter *expose* the underlying parameters such that users can adjust them before running the sorter. In fact, to ease the parameter adjustment process for a new user, SpikeInterface contains two functions, get_default_params() and get_params_description(), which return the developer-suggested value and description of each parameter, respectively. We believe that this functionality provides quick and accessible guidance to the users to make spike sorting *less* “black box” if anything. We would also like to point the editor and reviewers to a series of open analyses that showcase how SpikeInterface can be used for parameter sweeping and evaluation of spike sorting results

(https://spikeinterface.github.io/blog/example-of-parameters-optimization/). While we acknowledge that these aspects of SpikeInterface are important to highlight, we believe that they would require an extensive study to properly investigate so we decided to leave them out of the current manuscript. Our main point here is that this type of optimisation is very easy to perform with SpikeInterface.

2) The analyses and presentation of the comparison of spike sorters was viewed as weak. Reviewers agreed that careful manual curation is still the only right way to compare spike sorting results. Reviewers felt that it would be a mistake for readers who are new to the spike sorting process to look at the analyses shown in the paper as the right way to compare spike sorting results. Reviewers felt that a manual curation step is necessary to make the spike sorter evaluation process more useful.

In our initial manuscript, the presented analyses were designed to showcase the different ways that SpikeInterface could be used to compare spike sorters. We agree with the reviewers, however, that the shallowness of these analyses was a major weak point that needed to be addressed. Therefore, in our updated manuscript, we completely re-wrote the Results section. In this new Results section, we explore how combining the output of multiple spike sorters can be used to improve overall spike sorting performance, a method we call ensemble spike sorting. To this end, we first demonstrate that there is surprisingly low agreement among six spike sorters designed for high-count probes on a Neuropixels recording (analysis of other recordings are illustrated in the supplement). Next, we repeat this analysis with a simulated Neuropixels dataset (with known ground truth), finding that the sorter agreement level is comparable to that of real data and that almost all units that are found by only one sorter are actually false positive units. Finally, we ask two experts to manually curate the output of the real Neuropixels dataset (using Phy) and compare the manually curated datasets to a consensus sorting result (units agreed upon by at least two out of the six sorters). We find excellent agreement between the “good” units found in the manually curated datasets and the consensus sorting results. We believe that this new analysis is much more detailed and comprehensive and provides evidence that using multiple spike sorters can potentially inform (or even replace) the time consuming and subjective manual curation step in extracellular analysis pipelines.

3) It was suggested that the authors should attempt to perform some sort of smart cluster merging strategy that utilizes the output of different sorters.

As outlined above, our analysis now explores an ensemble-based approach to spike sorting which combines the output of several spike sorters to improve overall spike sorting performance. In addition, SpikeInterface now includes functionality to automatically generate this consensus set by merging the best unit matches among multiple sorters. While we found this can yield good results for ground truth data, we believe further analysis is required to validate this method. We are confident, however, that the agreement scores among multiple spike sorters can, at the very least, inform subsequent manual curation steps.

4) Another suggestion for a potentially useful addition was that users would be able to swap in and out different algorithmic components in the spike sorting pipeline. Combined with manual curation and comparison to "ground truth" data sets, reviewers felt that this could help users to determine the best algorithmic components for particular types of recordings.

We would like to point out (as expressed also by reviewer 1) that different processing steps in a spike sorting algorithm are not decoupled from each other. We therefore believe that swapping components between different algorithms cannot be a viable solution for building better spike sorting pipelines. Despite our reservations about the mix-and-match approach to spike sorting, we have added a module to the spiketoolkit package called sortingcomponents which will contain different algorithmic components for spike sorting.

Currently, this package just contains a decoupled detection algorithm, but we plan to add more functionality in future updates.

Reviewer #1:In this paper, the authors introduce a Python package for easily running many different spike sorters and exporting to many different formats. The goal is to make it easier for electrophysiologists to run their data through the spike sorters and output these results to Phy and other GUIs for data visualizations. While I agree that spike sorting is a hard problem and users need to be helped as much as possible, I don't think this framework helps a lot and will ultimately not find much use. I think Phy (already published and widely used) does most of the work that the authors suggest SpikeInterface should do, and in fact the main use case for SpikeInterface seems to be as an exporter to Phy. At its core, the code provided here is a set of file converters and code wrappers that further obfuscate the black-boxes that many spike sorters are, and make it more difficult for users to know how to build a successful spike sorting pipeline for their own data.

We thank the reviewer for the comments, however, we strongly disagree with this assessment. Phy is a tool for visualising and manually curating spike sorter outputs. In contrast, SpikeInterface aims to automate many aspects of a sorting pipeline, including file IO, pre and post-processing, running multiple spike sorting jobs, comparing spike sorting outputs, and computing quality metrics. As we show in our updated manuscript, the ability to efficiently run multiple spike sorters and to quickly compare their outputs offers new possibilities for automatic curation and unit annotation to inform manual curation. All analysis that can be performed with SpikeInterface is fully reproducible as well. We hope our revised paper better illustrates the many exciting use cases of SpikeInterface.

Reviewer #2:The work presented in this manuscript is of great interest to both spike sorting users and developers. The unified framework bridges the gap between the plethora of recording file formats and spike sorting packages, which is a major improvement in terms of spike sorting experience. The framework also features many interesting features related to spike sorting for processing recordings and sorting results. The manuscript is clearly written and introduces the functionality that is at hand in the framework in a concise way. Below is a list of major and minor comments that need to be addressed, however.1) Spikeinterface is portrayed as a general spike sorting framework. Still, the spike sorting workflow supported by spikeinterface appears to be geared towards specific kind of data and sorters, i.e. those that work on high electrode count continuous datasets. The authors should make explicit the assumptions that are made in spikeinterface regarding the data that is accepted (e.g. datasets with only waveform snippets appear not to be supported) and the minimal requirement for spike sorters (e.g. do spike sorters need to include their own spike detection algorithm and spike feature extraction?).

The reviewer correctly points out that datasets with only waveform snippets are not supported by SpikeInterface. Apart from this requirement, which is satisfied by most of modern acquisition systems, SpikeInterface does not make any other assumptions about the initial dataset. All supported spike sorters are end-to-end, i.e. they take a raw recording as input and output the sorted results. While many supported sorters are designed for high-count devices (e.g. IronClust, HerdingSpikes, Kilosort2), we also include sorters which are better suited for low-channel counts (Klusta, Wave-Clus, Mountainsort4). To help users better understand the different sorters supported by SpikeInterface, we added the get_sorte_description() function to the spikesorters package which provides a description of the algorithm, including its intended use.

2) The authors have chosen to run the spike sorters with their default parameters and without manual or automated refinement (i.e. noise cluster rejection, cluster merging/splitting). As many spike sorting algorithms explicitly depend on a manual cluster merging/splitting step after they have been applied to the data, it would be interesting to also provide an automated cluster merging (e.g., based on the ground truth as in Wouters, Kloosterman and Bertrand, 2019). This will improve the understanding of the true potential of a spike sorting algorithm, when comparing it to others in a ground-truth study. As a bare minimum, the authors should discuss the need of a post-sorting split/merge curation step and discuss the effect of leaving the step out on their results. Without such discussion, it would be premature to talk about a "consensus-based strategy" to select clusters (subsection “Application 1: Comparing Spike Sorters on Neuropixels Data”).

We thank the reviewer for this insightful comment which led us to completely re-write the Results section in our updated manuscript. In our new Results section, we provide evidence that a consensus-based spike sorting strategy is a viable alternative to using a single sorter. We demonstrate that units that are found by *only* one sorter mainly coincide with false positives (on simulated data). We also added a comparison between our consensus-based method and the manually curated outputs from two experts (on real data), showing that there is a large agreement between units labeled as “good” by the curators and the consensus sorting output. We believe that the new results in our updated manuscript provide initial evidence for the viability of a consensus-based strategy. In the Discussion section, we added a paragraph on Ensemble spike sorting to discuss the potential strengths and limitations of this strategy.

3) The authors define an agreement score to match clusters from different sorters and use the score to classify clusters (as compared to ground truth) as "well-detected", "false positive", "redundant" and "over-merged". However, a low agreement score could result from a high number of false positive detections or a high number of false negative detections (or both), and the interpretation would be different in these cases. In the extremes of no false positives or false negatives, an agreement score of 0.2 could either mean all spikes in a cluster represent 20% of the ground truth spikes (i.e. a clean partial cluster) or it could mean that all ground truth spikes represent 20% of the spikes in a cluster (i.e. a "dirty" over-merged cluster). Thus, the agreement score is not a good metric for the classification of the clusters. Instead, the authors should consider a classification based on different metric(s), e.g. both precision and recall.

We acknowledge that our proposed classification of sorted units is preliminary and cannot differentiate between some spike sorting failure modes. However, we still believe that it can provide insight into the strengths and weaknesses of different sorters despite not being a perfect solution. We agree that a more detailed analysis on multiple ground-truth recordings is needed to provide better classification rules. We address these concerns in the updated manuscript by adding the following sentence in the subsection “SpikeComparison”: "We would like to highlight to the reader that the unit classification proposed here is currently only based on agreement score (i.e. accuracy). More sophisticated classification rules could involve a combination of accuracy, precision, and recall values, which can be easily computed for each unit with the spikecomparison module."

4) We do not find the swarm plot in Figure 4 that compares the accuracy, precision and recall for multiple sorters very informative. First, the number of non-matched clusters is not obvious in this plot (we assume point with zero score are non-matched?). More importantly, there is often a trade-off between the number of false positive and false negatives, and each sorter may make a different trade-off, depending on the parameters. The swarm plot does not show the relation between precision and recall for each sorter, and a precision-recall scatter plot would be more informative.

To address this concern, we added a supplementary figure to the new manuscript that shows a scatter plot with precision vs recall for each sorter (Figure 2—figure supplement 1). As mentioned by reviewer 2, this figure helps reveal relevant differences among the chosen sorters and we thought it important to include in the new draft.

Reviewer #3:This submission describes a software toolbox aimed to facilitate the comparison of spike sorting algorithms. It is targeted for a broad user base, who may not have the time or technical ability to make such comparisons on their own. This tool addresses a need of the neuroscience community. Outlined below are number of suggested corrections.Introduction: Not all the listed sorters are truly fully-manual, ie Mclust is semiautomatic.

Thanks for catching this; We removed MClust from the list of fully manual spike sorters.

Subsection “Overview of SpikeInterface”: Roman numerals swapped for spikecomparison and spikewidgets.

We fixed the roman numerals accordingly.

Subsection “SpikeExtractors”: It is unclear how recordingextractor, a visualization tool, provides functionality required to excess data to evaluate the spike sorting pipeling. This becomes more clear later, but could be made more clear sooner.

A RecordingExtractor is not a visualization tool, but an file IO class that interfaces with the data. We are not sure what reviewer 3 means with this comment.

Subsection “SpikeExtractors” and subsection “SpikeToolkit”: The code snippets could be expanded to give more context and be more relevant.

In our updated manuscript, we improved the description before each code snippet to provide more context. We still chose to keep the code snippets minimal just to show the basic aspects of the API. The reader/user can find more detailed and comprehensive examples in the online documentation.

Subsection “Curation”: Instead of holding of for the future, this functionality would be nice to implement here, if it is not an unreasonable amount of work.

The curation module has now been extended to support curation based on all supported quality metrics.

Subsection “Using the Python API”: It could be said that spikeinterface is also handmade, maybe clarify the point.

The main point of that sentence is that, usually, custom scripts are not fully tested on continuous integration platforms, while SpikeInterface is. We believe that the paragraph conveys this message: "Unlike handmade scripts, SpikeInterface has a wide range of unit tests, employs continuous integration, and has been carefully developed by a team of researchers. Users, therefore, can have increased confidence that the pipelines they create are correct and reusable. Additionally, SpikeInterface tracks the entire provenance of the performed analysis, allowing other users (or the same user) to reproduce the analysis at a later date.".

Figure 3B is hard to read.

We agree with the reviewer and we removed the panel from the Figure.

Figure 3D, what are the color code agreement levels exactly, this is unclear.

The legend is shown on the top right and it indicates the number of k sorters that agree on a unit.

In Figure 4 it would be nice to see plotted SNR vs agreement score.

We added the suggested Figure as Figure 2—figure supplement 1B.

[Editors’ note: what follows is the authors’ response to the second round of review.]

Summary:The authors describe SpikeInterface, which is an integrated set of tools that makes it straightforward for researchers to set up a complete spike sorting workflow. SpikeInterface is modular (and extendible) and supports many common data formats and modern spike sorters. It provides postprocessing tools for characterization of the spike sorting results and for validation and comparison of multiple spike sorting results (e.g. against ground truth). SpikeInterface allows users to focus on the spike sorting results and curation, rather than having to glue together or (re)implement disparate tools themselves.Compared to the previous version, the authors have now more clearly outlined the goals of SpikeInterface in the introduction. In addition to comparing multiple sorters, they have added new results that indicate that combining the results of multiple spike sorters ("ensemble spike sorting") could help to reduce the number of false positive units, which is an interesting future direction that needs further investigation.Essential revisions:1) Regarding the ensemble spike sorting approach:

– From the results shown in Figure 3C, it seems that one won't need to run all 6 sorters to eliminate the false positives. Could the authors quantify the relative benefit of combining 2, 3 or more sorters over the use of a single sorter?

We have added Figure 3—figure supplement 1, and a brief passage of text to show how the detection of false/true positives depends on the number of sorters used, using the simulated ground-truth dataset. To this end, we tested all possible combinations of two to five sorters and quantified the fraction of identified false positives. The results show that already two sorters are sufficient to remove false positives, with little change when more sorters are added. However, the fraction of true positives in the ensemble sorting can increase substantially when more sorters are used. Therefore, the main benefit of combining multiple sorters is a more reliable identification of true positive units.

– One could imagine that combining the results of spike sorters that use a different class of algorithm would provide more benefit than combining two sorters that use the same algorithm. Do the authors observe this?

The sorters are highly variable in terms of performance which may occlude systematic differences between the main algorithms (template-based sorting and density-based clustering). We did observe a slightly better identification of false/true positives when different approaches (e.g. Kilosort2 and Ironclust) are combined, compared to the use of one approach alone (e.g. Kilosort2 and SpykingCircus). However, we found it difficult to judge if these biases were due to the central algorithm or due to other components of the sorting process. Therefore, we feel we could not give a well-qualified answer to this question as this would require a more systematic comparison of algorithms with equal pre/postprocessing and more ground-truth datasets with different generative assumptions.

– If reviewers understand correctly, many spike sorters will return a slightly different output when run a second time on the same data set. To what extent does running the same sorter twice give the same benefit (i.e. low agreement on false positive clusters) as running two different sorters?

We tested this phenomena for Kilosort2 and the simulated ground-truth recording from our manuscript. Each run returned slightly different results with different numbers of units each time. Using the same ensemble method, we found that all unmatched units were indeed false positives, but this removed only a subset of false positives (a certain fraction of false positives was always consistently found). A summary of the effect is shown in the following figure:

**Author response image 1. sa2fig1:** Comparison of five individual runs of Kilosort2 on the simulated Neuropixels recording. The top shows the proportions of units from each sorting found in k other sortings. Below these units are split according to false and true positive units after comparison to the ground-truth data. While a sizable fraction of false positive units are unique to each run of the sorter, many are identical in all sortings, indicating that variability in multiple sorter outputs cannot be used to reliably separate false and true positive units.

While potentially worth further investigation, we feel it is too preliminary to show these results as we do not understand where these differences originate and how they can be controlled. Therefore, we decided not to mention this result in the manuscript.

– Would using the ensemble spike sorting approach give similar results as using a more stringent selection of units found by a single sorter (e.g. based on cluster quality metrics, SNR, spike amplitude, etc.)?

We tested this idea using the most obvious quality measure, SNR, on a Kilosort2 sorting of the ground-truth data. We chose Kilosort2 since its performance on detecting true positive units is best among all the sorters. Here we found, surprisingly, that false positive units can have a large SNR so there is no obvious way to separate these. This is illustrated in Figure 3—figure supplement 2, and a short passage was added to this effect.

2) The authors used an arbitrary agreement score threshold of 0.5, which they acknowledge is a pragmatic but not necessarily the best choice to match units found by multiple sorters. Reviewers did not think it is necessary to change the way units are matched, but to provide more insight into the matching process it would be helpful to know what the distribution of unit agreement scores for matching pairs looks like.

We added a histogram of agreement scores for the ground-truth data (Figure 1—figure supplement 2), which shows that the majority of matches have scores >0.8.

3) When evaluating the spike sorting results on the simulated data set, the authors only mention matches and false positives. Reviewers did not see a mention of the number of false negatives (i.e. number of ground units that were missed by the sorters). Could the authors also indicate to what extent spikes of the missed clusters actually show up as part of the false positive units (e.g. because false positive units are actually overly split/merged true units)?

There are 250 ground-truth units, which we now indicate as an horizontal dashed line in Figure 2E.